# DiReCT: Diagnostic Reasoning for Clinical Notes via Large Language Models

**Bowen Wang**♣◇,∗ **Jiuyang Chang**♡,∗ **Yiming Qian**♠,† **Guoxin Chen**★, **Junhao Chen**◇,
**Zhouqiang Jiang**◇, **Jiahao Zhang**◇, **Yuta Nakashima**◇♣, **Hajime Nagahara**◇♣

♣Premium Research Institute for Human Metaverse Medicine (WPI-PRIMe), Osaka University,
♡Department of Cardiology, The First Affiliated Hospital of Dalian Medical University,
★Institute of Computing Technology, Chinese Academy of Science
◇D3 Center, Osaka University, ♠Agency for Science, Technology and Research (A*STAR),
{wang, n-yuta, nagahara}@ids.osaka-u.ac.jp
changjiuyang@firsthosp-dmu.com
qiany@ihpc.a-star.edu.sg, gx.chen.chn@gmail.com
{junhao, zhouqiang, jiahao}@is.ids.osaka-u.ac.jp

## Abstract

Large language models (LLMs) have recently showcased remarkable capabilities, spanning a wide range of tasks and applications, including those in the medical domain. Models like GPT-4 excel in medical question answering but may face challenges in the lack of interpretability when handling complex tasks in real clinical settings. We thus introduce the diagnostic reasoning dataset for clinical notes (DiReCT), aiming at evaluating the reasoning ability and interpretability of LLMs compared to human doctors. It contains 511 clinical notes, each meticulously annotated by physicians, detailing the diagnostic reasoning process from observations in a clinical note to the final diagnosis. Additionally, a diagnostic knowledge graph is provided to offer essential knowledge for reasoning, which may not be covered in the training data of existing LLMs. Evaluations of leading LLMs on DiReCT bring out a significant gap between their reasoning ability and that of human doctors, highlighting the critical need for models that can reason effectively in real-world clinical scenarios [‡].

## 1 Introduction

Recent advancements of large language models (LLMs) [Zhao et al., 2023] have ushered in new possibilities and challenges for a wide range of natural language processing (NLP) tasks [Min et al., 2023]. In the medical domain, these models have demonstrated remarkable prowess [Anil et al., 2023, Han et al., 2023], particularly in medical question answering (QA) [Jin et al., 2021]. Leading-edge models, such as GPT-4 [OpenAI, 2023a], exhibit profound proficiency in understanding and generating text [Bubeck et al., 2023], even achieved high scores on the United States Medical Licensing Examination (USMLE) questions [Nori et al., 2023].

Despite the advancements, interpretability is critical, particularly in medical NLP tasks [Liévin et al., 2024] because these tasks directly impact patient health and treatment decisions. Without clear interpretability, there's a risk of misdiagnosis and improper treatment, making it vital for ensuring medical safety. Some studies assess this capability over medical QA [Pal et al., 2022, Li et al., 2023,

---

[∗]Equal contribution.
[†]Corresponding author.
[‡]Data and code are available at https://github.com/wbw520/DiReCT.

38th Conference on Neural Information Processing Systems (NeurIPS 2024) Track on Datasets and Benchmarks.

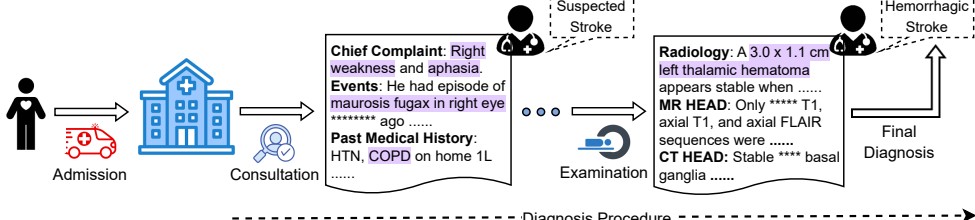

Figure 1: When a patient is admitted, an initial consultation takes place to collect subjective information. Subsequent observations may then require further examination to confirm the diagnosis.

Chen et al., 2024] or natural language inference (NLI) [Jullien et al., 2023]. Putting more attention on interpretability, they use relatively simple tasks as testbeds, taking short text as input. Nevertheless, real-world clinical tasks are often more complex [Gao et al., 2023a], as illustrated in Figure 1, a typical diagnosis requires comprehending and combining various information, such as health records, physical examinations, and laboratory tests, for further reasoning of possible diseases in a step-by-step manner following the established guidelines. This observation suggests that both *perception*, or reading (e.g., finding necessary information in the medical record) and *reasoning* (determining the disease based on the observations) should be counted when evaluating interpretability in LLM-based medical NLP tasks.

For a more comprehensive evaluation of LLMs for supporting diagnosis in a more realistic setting, we propose a **Di**agnostic **Re**asoning dataset for **C**linical no**T**es (DiReCT). The task basically is predicting the diagnosis from a *clinical note* of a patient, which is a collection of various medical records, written in natural language. Our dataset contains 511 clinical notes spanning 25 disease categories, sampled from a publicly available database, MIMIC-IV [Johnson et al., 2023]. Each clinical note undergoes fine-grained annotation by professional physicians. The annotators (i.e., the physicians) are responsible for identifying the text, or the *observation*, in the note that leads to a certain diagnosis, as well as the explanation. The dataset also provides a diagnostic knowledge graph based on existing diagnostic guidelines to facilitate more consistent annotations and to supply a model with essential knowledge for reasoning that might not be encompassed in its training data.

To underscore the challenge offered by our dataset, we propose a simple AI-agent based baseline [Xi et al., 2023, Tang et al., 2023] that utilizes the knowledge graph to decompose the diagnosis into a sequence of diagnoses from a smaller number of observations. Our experimental findings indicate that current state-of-the-art LLMs still fall short of aligning well with human doctors.

**Contribution**. DiReCT offers a new challenge in diagnosis from a complex clinical note with explicit knowledge of established guidelines. This challenge aligns with a realistic medical scenario that doctors are experiencing. In the application aspect, the dataset facilitates the development of a model to support doctors in diagnosis, which is error-prone [Middleton et al., 2013, Liu et al., 2022]. From the technical aspect, the dataset can benchmark models' ability to read long text and find necessary observations for *multi-evidence entailment tree* reasoning, an extension of the original entailment tree explanation [Dalvi et al., 2021] for complex scenarios in medical NLP tasks. As shown in Figure 3, this is not trivial because of the variations in writing; superficial matching does not help, and medical knowledge is vital. Meanwhile, reasoning itself is facilitated by the knowledge graph. The model does not necessarily have the knowledge of diagnostic guidelines. With this choice, the knowledge graph explains the reasoning process, which is also beneficial when deploying such a diagnosis assistant system in practical uses.

## 2 Related Works

**Natural language explanation**. Recent advancements in NLP have led to significant achievements [Min et al., 2023]. However, existing models often lack explainability, posing potential risks [Danilevsky et al., 2020, Gurrapu et al., 2023]. Numerous efforts have been made to address this challenge. One effective approach is to provide a human-understandable *plain text* explanation alongside the model's output [Camburu et al., 2018, Rajani et al., 2019]. Another strategy involves identifying *evidence* within the input that serves as a rationale for the model's decisions, aligning with

Table 1: Comparison of existing datasets for medical reasoning tasks and ours. "t" and "w" mean tokens and words for the length of input, respectively.

| Dataset | Task | Data Source | Length | Explanation | # Cases |
|---|---|---|---|---|---|
| MedMCQA [Pal et al., 2022] | QA | Examination | 9.93 t | Plain Text | 194,000 |
| ExplainCPE [Li et al., 2023] | QA | Examination | 37.79 w | Plain Text | 7,000 |
| JAMA Challenge [Chen et al., 2024] | QA | Clinical Cases | 371 w | Plain Text | 1,524 |
| Medbullets [Chen et al., 2024] | QA | Online Questions | 163 w | Plain Text | 308 |
| N2N2 [Gao et al., 2022] | Sum | Clinical Notes | 785.46 t | Evidences | 768 |
| NLI4CT [Jullien et al., 2023] | NLI | Clinical Trail Reports | 10-35 t | Multi-hop | 2,400 |
| NEJM CPC [Zack et al., 2023] | CD | Clinical Cases | - | Plain Text | 2,525 |
| DiReCT (Ours) | CD | Clinical Notes | 1074.6 t | Entailment Tree | 511 |

human reasoning [DeYoung et al., 2020]. Expanding on this concept, [Jhamtani and Clark, 2020] introduces chain-structured explanations, given that a diagnosis can demand multi-hop reasoning. This idea is further refined by ProofWriter [Tafjord et al., 2021] through a proof stage for explanations, and by [Zhao et al., 2021] through retrieval from a corpus. [Dalvi et al., 2021] proposes the *entailment tree*, offering more detailed explanations and facilitating inspection of the model's reasoning. More recently, [Zhang et al., 2024] employed cumulative reasoning to tap into the potential of LLMs to provide explanation via a *directed acyclic graph*. Although substantial progress has been made, interpreting NLP tasks in medical domains remains an ongoing challenge [Liévin et al., 2024].

**Benchmarks of interpretability in the medical domain** Several datasets are designed to assess a model's reasoning together with its interpretability in medical NLP (Table 1). MedMCQA [Pal et al., 2022] and other medical QA datasets [Li et al., 2023, Chen et al., 2024] provide plain text as explanations for QA tasks. NLI4CT [Jullien et al., 2023] uses clinical trial reports, focusing on NLI supported by multi-hop reasoning. N2N2 [Gao et al., 2022] proposes a summarization (Sum) task for a diagnosis based on multiple pieces of evidence in the input clinical note. NEJM CPC [Zack et al., 2023] interprets clinicians' diagnostic reasoning as plain text for reasoning clinical diagnosis (CD). DR.BENCH [Gao et al., 2023b] aggregates publicly available datasets to assess the diagnostic reasoning of LLMs. Utilizing an multi-evidence entailment tree explanation, DiReCT introduces a more rigorous task to assess whether LLMs can align with doctors' reasoning in real clinical settings.

## 3   A benchmark for Clinical Notes Diagnosis

This section first detail clinical notes (Section 3.1). We also describes the knowledge graph that encodes existing guidelines (Section 3.2). Our task definition, which tasks a clinical note and the knowledge graph as input is given in Section 3.4. We then present our annotation process for clinical notes (Section 3.3) and the evaluation metrics (Section 3.5).

### 3.1   Clinical Notes

Clinical notes used in DiReCT are stored in the SOAP format [Weed, 1970]. A clinical note comprises four components: In the *subjective* section, the physician records the patient's chief complaint, the history of present illness, and other subjective experiences reported by the patient. The *objective* section contains structural data obtained through examinations (inspection, auscultation, etc.) and other measurable means. The *assessment* section involves the physician's analysis and evaluation of the patient's condition. This may include a summary of current status, *etc*. Finally, the *plan* section outlines the physician's proposed treatment and management plan. This may include prescribed medications, recommended therapies, and further investigations. A clinical note also includes a primary discharge diagnosis (PDD) in the assessment section.

DiReCT's clinical notes are sourced from the MIMIC-IV dataset [Johnson et al., 2023] (PhysioNet Credentialed Health Data License 1.5.0), which encompasses over 40,000 patients admitted to the intensive care units. Each note contains clinical data for a patient. To construct DiReCT, we curated a subset of 511 notes whose PDDs fell within one of 25 disease categories $i$ in 5 medical domains.

In our task, a note $R = \{r\}$ is an excerpt of 6 clinical data in the subjective and objective sections (i.e., $|R| = 6$): chief complaint, history of present illness, past medical history, family history, physical

exam, and pertinent results.[1] We also identified the PDD $d^\star$ associated with $R$.[2] The set of $d^\star$'s for all $R$'s collectively forms $\mathcal{D}^\star$. We manually removed any descriptions that disclose the PDD in $R$.

## 3.2 Diagnostic Knowledge Graph

Existing knowledge graphs for the medical domain, e.g., UMLS KG [Bodenreider, 2004], lack the ability to provide specific clinical decision support (e.g., diagnostic threshold, context-specific data, dosage information, etc.), which are critical for accurate diagnosis.

Our knowledge graphs $\mathcal{K} = \{k_i\}$ is a collection of graph $k_i$ for disease category $i$. $k_i$ is based on the diagnosis criteria in existing guidelines (refer to supplementary material for details). $k_i$'s nodes are either premise $p \in \mathcal{P}_i$ (medical statement, e.g., `Headache is a symptom of`) and diagnoses $d \in \mathcal{D}_i$ (e.g., `Suspected Stroke`). $k_i$ consists of two different types of edges. One is *premise-to-diagnosis* edges $\mathcal{S}_i = \{(p, d)\}$; an edge is from $p$ to $d$. This edge represents the necessary premise $p$ to make a diagnosis $d$. We refer to them as *supporting* edges. The other is *diagnosis-to-diagnosis* edges $\mathcal{F}_i = \{(d, d')\}$, where $d, d' \in \mathcal{D}_i$ and the edge is from $d$ to $d'$, which represents the diagnostic flow. These edges are referred to as *procedural* edges.

A disease category is defined according to an existing guideline, which starts from a certain diagnosis; therefore, a procedural graph $g_i = (\mathcal{D}_i, \mathcal{F}_i)$ ($\mathcal{G} = \{g_i\}$) has only one root node and arbitrarily branches toward multiple leaf nodes that represent PDDs (i.e., the clinical notes in DiReCT are chosen to cover all leaf nodes of $g_i$). Thus, $g_i$ is a *tree*. We denote the set of the leaf nodes (or PDDs) as $\mathcal{D}_i^\star \subset \mathcal{D}_i$. The knowledge graph is denoted by $k_i = (\mathcal{D}_i, \mathcal{P}_i, \mathcal{S}_i, \mathcal{F}_i)$.

Figure 2 shows a part of $k_i$, where $i$ is `Acute Coronary Syndromes (ACS)`. Premises in $\mathcal{P}_i$ and diagnoses in $\mathcal{D}_i$ are given in the blue and gray boxes, while PDDs in $\mathcal{D}_i^\star$ are ones without outgoing edges (i.e., `STEMI-ACS` and `NSTEMI-ACS`, and `UA`). The black and red arrows are edges in $\mathcal{S}$ and $\mathcal{F}$, respectively, where the black arrows indicate the supporting edges.

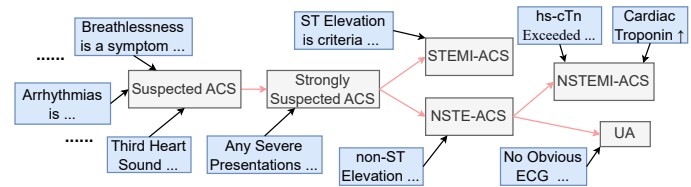

Figure 2: A part of $k_i$ for $i$ being `Acute Coronary Syndromes`.

$\mathcal{K}$ serves two essential functions: (1) They serve as the gold standard for annotation, guiding doctors in the precise and uniform interpretation of clinical notes. (2) Our task also allows a model to use them to ensure the output from an LLM can be closely aligned with the reasoning processes of medical professionals.

## 3.3 Data Annotation

Let $d^\star \in \mathcal{D}_i^\star$ denote the PDD of disease category $i$ associated with $R$. We can find a subgraph $k_i(d^\star)$ of $k_i$ that contains all ancestors of $d^\star$, including premises in $\mathcal{P}_i$. We also denote the set of supporting edges in $k_i(d^\star)$ as $\mathcal{S}_i(d^\star)$. Our annotation process is, for each supporting edge $(p, d) \in \mathcal{S}_i(d^\star)$, to extract observation $o \in \mathcal{O}$ in $R$ (highlighted text in the clinical note in Figure 3) and provide rationalization $z$ of this *deduction* why $o$ is a support for $d$ or corresponds to $p$.[3] They form the explanation $\mathcal{E} = \{(o, z, d)\}$ for $(R, d^\star)$. This annotation process was carried out by 9 clinical physicians and subsequently verified for accuracy and completeness by three senior medical experts.

Table 2 summarizes statistics of our dataset. The second and third columns ("# cats." and "# samples") show the numbers of disease categories and samples in the respective medical domains. $|\mathcal{D}_i|$ and $|\mathcal{D}_i^\star|$ are the total numbers of diagnoses (diseases) and PDDs, summed over all diagnostic categories

---

[1]We excluded data, such as review system and social history, because they are often missing in the original clinical notes and are less relevant to the diagnosis.

[2]All clinical notes in DiReCT are related to only one PDD, and there is no secondary discharge diagnosis.

[3]All annotations strictly follow the procedural flow in $k_i$, and each observation is only related to one diagnostic node. If $R$ does not provide sufficient observations for the PDD (which may happen when a certain test is omitted), the annotators were asked to add plausible observations to $R$. Refer to amended data points in supplementary for details.

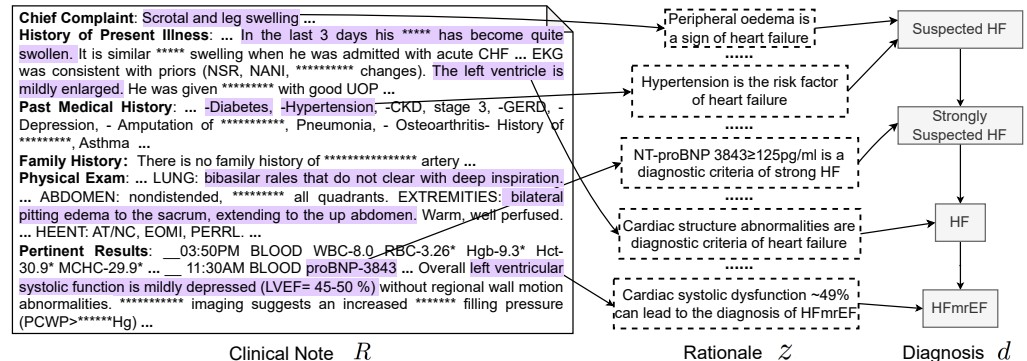

Figure 3: An annotation sample of `Heart Failure` (HF). The left part is the clinical note alongside extracted observations by a doctor. The middle part outlines the steps of the rationale for the premise corresponding to each diagnostic node shown in the right part.

in the medical domain, respectively. $|\mathcal{O}|$ is the average number of annotated observations. "Length" is the average number of tokens in $R$.

### 3.4 Task Definition

We propose two tasks with different levels of supplied external knowledge. The first task is, given $R$ and $\mathcal{G}$, to predict the associated PDD $d^\star$ and generate an explanation $\mathcal{E}$ that explains the model's diagnostic procedure from $R$ to $d^\star$, i.e., letting $M$ denote a model:

$$\hat{d}^\star, \hat{\mathcal{E}} = M(R, \mathcal{G}), \qquad (1)$$

Table 2: Statistics of DiReCT.

| Medical domain | # cat. | # samples | $|\mathcal{D}_i|$ | $|\mathcal{D}_i^\star|$ | $|\mathcal{O}|$ | Length |
|---|---|---|---|---|---|---|
| Cardiology | 7 | 184 | 27 | 16 | 8.7 | 1156.6 t |
| Gastroenterology | 4 | 103 | 11 | 7 | 4.3 | 1026.0 t |
| Neurology | 5 | 77 | 17 | 11 | 11.9 | 1186.3 t |
| Pulmonology | 5 | 92 | 26 | 17 | 10.7 | 940.7 t |
| Endocrinology | 4 | 55 | 20 | 14 | 6.9 | 1063.5 t |
| Overall | 25 | 511 | 101 | 65 | 8.5 | 1074.6 t |

where $\hat{d}^\star \in \cup_i \mathcal{D}_i^\star$ and $\hat{\mathcal{E}}$ are predictions for the PDD and explanation, respectively. With this task, the knowledge of specific diagnostic procedures in existing guidelines can be used for prediction, facilitating interpretability. The second task takes $\mathcal{K}$ as input instead of $\mathcal{G}$, i.e.,:

$$\hat{d}^\star, \hat{\mathcal{E}} = M(R, \mathcal{K}). \qquad (2)$$

This task allows for the use of broader knowledge of premises for prediction. One may also try a task without any external knowledge.

### 3.5 Evaluation Metrics

We designed three metrics to quantify the predictive performance over our benchmark.

(1) *Accuracy of diagnosis* $Acc^{\text{diag}}$ evaluates if a model can correctly identify the diagnosis. $Acc^{\text{diag}} = 1$ if $d^\star = \hat{d}$, and $Acc^{\text{diag}} = 0$ otherwise. The average is reported.

(2) *Completeness of observations* $Obs^{\text{comp}}$ evaluates whether a model extracts all and only necessary observations for the prediction. Let $\mathcal{O}$ and $\hat{\mathcal{O}}$ denote the sets of observations in $\mathcal{E}$ and $\hat{\mathcal{E}}$, respectively. The metric is defined as $Obs^{\text{comp}} = |\mathcal{O} \cap \hat{\mathcal{O}}|/|\mathcal{O} \cup \hat{\mathcal{O}}|$, where the numerator is the number of observations that are common in both $\mathcal{O}$ and $\hat{\mathcal{O}}$.[4] This metric simultaneously evaluates the correctness of each observation and the coverage. To supplement it, we also report the precision $Obs^{\text{pre}}$ and recall $Obs^{\text{rec}}$, given by $Obs^{\text{pre}} = |\mathcal{O} \cap \hat{\mathcal{O}}|/|\hat{\mathcal{O}}|$ and $Obs^{\text{rec}} = |\mathcal{O} \cap \hat{\mathcal{O}}|/|\mathcal{O}|$.

(3) *Faithfulness of explanations* evaluates if the diagnostic flow toward the PDD is fully supported by observations with faithful rationalizations. This involves establishing a one-to-one correspondence between deductions in the prediction and the ground truth. We use the correspondences established for computing $Obs^{\text{comp}}$. Let $o \in \mathcal{O}$ and $\hat{o} \in \hat{\mathcal{O}}$ denote corresponding observations. This correspondence

---

[4] We find the common observations with an LLM (refer to the supplementary material for more detail).

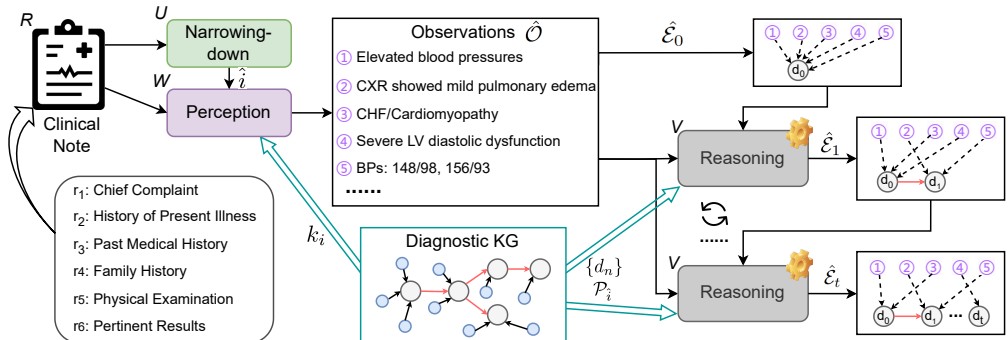

Figure 4: Pipeline of our baseline. The dotted line in the right-most boxes means deductions from an observation to a diagnosis.

is considered successful if $z$ and $\hat{z}$ as well as $d$ and $\hat{d}$ associated with $o$ and $\hat{o}$ matches. Let $m(\mathcal{E}, \hat{\mathcal{E}})$ denote the number of successful matches. We use the ratio of $m(\mathcal{E}, \hat{\mathcal{E}})$ to $|\mathcal{O} \cap \hat{\mathcal{O}}|$ and $|\mathcal{O} \cup \hat{\mathcal{O}}|$ as evaluation metrics $Exp^{\mathrm{com}}$ and $Exp^{\mathrm{all}}$, respectively, to see failures come from observations or explanations and diagnosis.

## 4   Baseline

Figure 4 provides an overview of our baseline, which comprises three LLM-based modules: narrowing-down ($U$), perception ($W$), and reasoning ($V$). In our experiments, each module utilizes the same type of LLM with different prompts (refer to the supplementary material for more details). $U$ analyze the entire note $R$ to determine the possible disease type $\hat{i}$. $W$ extracts observations that may lead to diseases from each $r$, producing a list of original disease descriptions. $V$ iteratively derives possible diseases from observations based on the diagnosis knowledge graph, providing rationales for each deduction $(o, z, d)$.

The narrowing-down module $U$ takes $R$ as input to make a prediction $\hat{i}$ of the disease category, i.e., $\hat{i} = U(R)$. Let $d_t \in \mathcal{D}_{\hat{i}}$ be the diagnosis that has been reached with $t$ iterations over $k_{\hat{i}}$, where $t$ corresponds to the depth of node $d_t$ and so is less than or equal to the depth of $k_{\hat{i}}$. $d_0$ is the root node of $k_{\hat{i}}$. For $d_0$, we apply the perception module to extract all observations in $R$ and explanation $\mathcal{E}_0$ to support $d_0$ as

$$\hat{\mathcal{O}}, \hat{\mathcal{E}}_0 = W(d_0, k_{\hat{i}}). \tag{3}$$

$k_{\hat{i}}$ is supplied to facilitate the model to extract all observations for the following reasoning process.[5] After the perception module $W$ (iteration $t = 0$), we obtain all observations $\hat{\mathcal{O}}$, the root node of the diagnosis $d_0$, and an explanation $\hat{\mathcal{E}}_0$ for the initial iteration. Assuming that by iteration $t$, we already know the diagnosis for iteration $t$ as $d_t$. $\{d_n\}$ is the set of $d_t$'s children, and $\mathcal{P}_{\hat{i}}(\{d_n\})$ represents the corresponding premises that support each $d_n$. $V$ identifies the diagnosis for the next step, $d_{t+1}$, and provides a justification $\mathcal{E}_{t+1}$. $V$ will verify if there is any $\hat{o}$ in $\hat{\mathcal{O}}$ that supports a $d_n$. If fully supported, $d_n$ is identified as $d_{t+1}$ for the $(t+1)$-th iteration, i.e.,

$$d_{t+1}, \hat{\mathcal{E}}_{t+1} = V(\hat{\mathcal{O}}, \{d_n\}, \mathcal{P}_{\hat{i}}(\{d_n\})), \tag{4}$$

$V$ continues until $d_{t+1}$ in $\mathcal{D}^*$ is identified. If no observation supports a $d_n$, the reasoning process will be stopped.

In our annotation, an observation $o$ is associated with only one $d$. However, our method employs an iterative reasoning pipeline. Initially, the perception module $W$ generates an explanation set $\hat{\mathcal{E}}_0$, linking all $\hat{o}$ to $d_0$. During the $t$-th iteration of $V$, the explanation set is $\hat{\mathcal{E}}_t$, where at least one $\hat{o}$ is

---

[5]We used only pairs of an observation and a premise. We abuse $\mathcal{K}$ to mean this for notation simplicity. The perception model can also utilize $g_i$ instead of $k_i$ for the first task.

Table 3: Evaluation of diagnostic reasoning ability using $\mathcal{G}$ or $\mathcal{K}$ as input.

| Task | Models | Diagnosis | | Observation | | | Explanation | |
|---|---|---|---|---|---|---|---|---|
| | | $Acc^{\text{cat}}$ | $Acc^{\text{diag}}$ | $Obs^{\text{pre}}$ | $Obs^{\text{rec}}$ | $Obs^{\text{comp}}$ | $Exp^{\text{com}}$ | $Exp^{\text{all}}$ |
| With $\mathcal{G}$ | Zephyr 7B | 0.274 | 0.151 | $0.123_{\pm 0.200}$ | $0.115_{\pm 0.166}$ | $0.092_{\pm 0.108}$ | $0.071_{\pm 0.139}$ | $0.014_{\pm 0.037}$ |
| | Mistral 7B | 0.507 | 0.306 | $0.211_{\pm 0.190}$ | $0.317_{\pm 0.253}$ | $0.173_{\pm 0.157}$ | $0.230_{\pm 0.312}$ | $0.062_{\pm 0.088}$ |
| | Mixtral 8×7B | 0.413 | 0.237 | $0.147_{\pm 0.165}$ | $0.266_{\pm 0.261}$ | $0.124_{\pm 0.138}$ | $0.144_{\pm 0.268}$ | $0.029_{\pm 0.056}$ |
| | LLama3 8B | 0.569 | 0.364 | $0.248_{\pm 0.157}$ | $0.410_{\pm 0.218}$ | $0.211_{\pm 0.138}$ | $0.325_{\pm 0.375}$ | $0.087_{\pm 0.118}$ |
| | LLama3 70B | 0.822 | 0.606 | $0.306_{\pm 0.151}$ | $\mathbf{0.543_{\pm 0.183}}$ | $0.279_{\pm 0.146}$ | $0.409_{\pm 0.328}$ | $0.124_{\pm 0.120}$ |
| | GPT-3.5 turbo | 0.679 | 0.455 | $0.389_{\pm 0.212}$ | $0.351_{\pm 0.192}$ | $0.275_{\pm 0.167}$ | $0.331_{\pm 0.366}$ | $0.103_{\pm 0.127}$ |
| | GPT-4 turbo | **0.804** | **0.610** | $\mathbf{0.486_{\pm 0.207}}$ | $0.481_{\pm 0.180}$ | $\mathbf{0.391_{\pm 0.189}}$ | $\mathbf{0.481_{\pm 0.362}}$ | $\mathbf{0.210_{\pm 0.188}}$ |
| With $\mathcal{K}$ | LLama3 8B | 0.576 | 0.344 | $0.235_{\pm 0.162}$ | $0.394_{\pm 0.227}$ | $0.199_{\pm 0.142}$ | $0.327_{\pm 0.375}$ | $0.087_{\pm 0.114}$ |
| | LLama3 70B | 0.786 | 0.652 | $0.268_{\pm 0.147}$ | $\mathbf{0.524_{\pm 0.211}}$ | $0.258_{\pm 0.142}$ | $0.549_{\pm 0.372}$ | $0.152_{\pm 0.130}$ |
| | GPT-3.5 turbo | 0.652 | 0.413 | $0.347_{\pm 0.241}$ | $0.279_{\pm 0.203}$ | $0.232_{\pm 0.184}$ | $0.374_{\pm 0.408}$ | $0.121_{\pm 0.152}$ |
| | GPT-4 turbo | **0.808** | **0.611** | $\mathbf{0.470_{\pm 0.209}}$ | $0.459_{\pm 0.190}$ | $\mathbf{0.371_{\pm 0.192}}$ | $\mathbf{0.645_{\pm 0.385}}$ | $\mathbf{0.273_{\pm 0.216}}$ |

linked to $d_t$. The final diagnosis explanation is the combination of $\hat{\mathcal{E}}_0, \ldots, \hat{\mathcal{E}}_T$ and $d_0, \ldots, d_T$, where $T$ represents the final iteration. In this combination, if an $\hat{o}$ is eventually processed in the iteration for $\hat{\mathcal{E}}_t$, the corresponding $(o, z, d)$ in all preceding $\hat{\mathcal{E}}_0, \ldots, \hat{\mathcal{E}}_{t-1}$ will be removed. That is, $\hat{o}$ will always be possessed by the $d_t$ closest to the leaf PDD node.

# 5 Experiments

## 5.1 Experimental Setup

We assess the reasoning capabilities of 7 recent LLMs from diverse families and model sizes, including 5 instruction-tuned models that are openly accessible: LLama3 8B and 70B [AI@Meta, 2024], Zephyr 7B [Tunstall et al., 2023], Mistral 7B [Jiang et al., 2023], and Mixtral 8×7B [Jiang et al., 2023]. We have also obtained access to private versions of the GPT-3.5 turbo [OpenAI, 2023b] and GPT-4 turbo [OpenAI, 2023a] [6], which are high-performance closed-source models. Each LLM is utilized to implement our baseline's narrowing-down, perception, and reasoning modules. The temperature is set to 0. For computing evaluation metrics, we use LLama3 8B with few-shot prompts to make correspondences between $\mathcal{O}$ and $\hat{\mathcal{O}}$ as well as to verify a match between predicted and ground-truth explanations (refer to the supplementary material for more details).

## 5.2 Results

**Comparison among LLMs.** Table 3 shows the performance of our baseline built on top of various LLMs. We first evaluate a variant of our task that takes graph $\mathcal{G} = \{\mathcal{G}_i\}$ consisting of only procedural flow as external knowledge instead of $\mathcal{K}$. Comparison between $\mathcal{G}$ and $\mathcal{K}$ demonstrates the importance of supplying premises with the model and LLMs' capability to make use of extensive external knowledge that may be superficially different from statements in $R$. Subsequently, some models are evaluated with our task using $\mathcal{K}$. In addition to the metrics in Section 3.5, we also adopt the *accuracy of disease category $Acc^{\text{cat}}$*, which gives 1 when $\hat{i} = i$, as our baseline's performance depends on it.

With $\mathcal{G}$, we can see that GPT-4 achieves the best performance in most metrics, especially related to observations and explanations, surpassing LLama3 70B by a large margin. In terms of accuracy (in both category and diagnosis levels), LLama3 70B is comparable to GPT-4. LLama3 70B also has a higher $Obs^{\text{rec}}$ but low $Obs^{\text{pre}}$ and $Obs^{\text{comp}}$, which means that this model tends to extract many observations. Models with high diagnostic accuracy are not necessarily excel in finding essential information in long text (i.e., observations) and generating reasons (i.e., explanations).

When $\mathcal{K}$ is given, all models show better diagnostic accuracy (in LLama3 70B) and explanations, while observations are slightly degraded (this may related to the instruction following ability due to the input length when giving $\mathcal{K}$ as input). GPT-4 with $\mathcal{K}$ enhances $Acc^{\text{diag}}$, $Exp^{\text{com}}$, and $Exp^{\text{all}}$ scores. This suggests that premises and supporting edges are beneficial for diagnosis and explanation. Lower

---

[6]These two models are housed on a HIPPA-compliant instance within Microsoft Azure AI Studio. No data is transferred to either Microsoft or OpenAI. This secure environment enables us to safely conduct experiments with the MIMIC-IV dataset, in compliance with the Data Use Agreement.

Table 4: Evaluation of diagnostic reasoning ability without external knowledge.

| Task | Models | $Acc^{\text{diag}}$ | Observation | | | Explanation | |
|------|--------|------|---------------------|---------------------|----------------------|----------------------|----------------------|
| | | | $Obs^{\text{pre}}$ | $Obs^{\text{rec}}$ | $Obs^{\text{comp}}$ | $Exp^{\text{com}}$ | $Exp^{\text{all}}$ |
| With $\mathcal{D}^\star$ | LLama3 8B | 0.070 | $0.154_{\pm0.139}$ | $0.330_{\pm0.244}$ | $0.135_{\pm0.122}$ | $0.020_{\pm0.100}$ | $0.004_{\pm0.016}$ |
| | LLama3 70B | 0.502 | $0.257_{\pm0.150}$ | $\mathbf{0.509_{\pm0.213}}$ | $0.237_{\pm0.145}$ | $0.138_{\pm0.209}$ | $0.034_{\pm0.054}$ |
| | GPT-3.5 turbo | 0.223 | $0.164_{\pm0.242}$ | $0.149_{\pm0.212}$ | $0.116_{\pm0.174}$ | $0.091_{\pm0.231}$ | $0.025_{\pm0.065}$ |
| | GPT-4 turbo | $\mathbf{0.636}$ | $\mathbf{0.461_{\pm0.206}}$ | $0.482_{\pm0.160}$ | $\mathbf{0.378_{\pm0.174}}$ | $\mathbf{0.186_{\pm0.221}}$ | $\mathbf{0.074_{\pm0.090}}$ |
| No Knowledge | LLama3 8B | 0.023 | $0.137_{\pm0.159}$ | $0.258_{\pm0.274}$ | $0.119_{\pm0.141}$ | $0.018_{\pm0.083}$ | $0.006_{\pm0.026}$ |
| | LLama3 70B | 0.037 | $0.246_{\pm0.148}$ | $\mathbf{0.504_{\pm0.222}}$ | $0.227_{\pm0.148}$ | $0.022_{\pm0.093}$ | $0.007_{\pm0.030}$ |
| | GPT-3.5 turbo | 0.059 | $0.161_{\pm0.238}$ | $0.148_{\pm0.215}$ | $0.113_{\pm0.171}$ | $0.036_{\pm0.131}$ | $0.011_{\pm0.039}$ |
| | GPT-4 turbo | $\mathbf{0.074}$ | $\mathbf{0.410_{\pm0.208}}$ | $0.443_{\pm0.191}$ | $\mathbf{0.324_{\pm0.182}}$ | $\mathbf{0.047_{\pm0.143}}$ | $\mathbf{0.019_{\pm0.058}}$ |

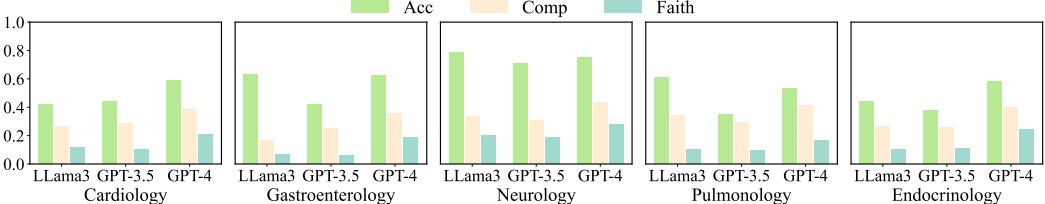

Figure 5: Performance of LLama3 70B, GPT-3.5, and GPT-4 under different medical domains. We use the task with $\mathcal{G}$.

observational performance may indicate that the models lack the ability to associate premises and text in $R$, which are often superficially different though semantically consistent.

LLMs may undergo inherent challenges for evaluation when no external knowledge is supplied. They may have the knowledge to diagnose but cannot make consistent observations and explanations that our task expects through $\mathcal{K}$. To explore this, we evaluate two settings: (1) giving $D^\star$ and (2) no knowledge is supplied to a model (shown in Table 4). The prompts used for this setup are detailed in the supplementary material. We do not evaluate the accuracy of disease category prediction as it is basically the same as Table 3. We can clearly see that with $\mathcal{D}^\star$, GPT-4's diagnostic and observational scores are comparable to those of the task with $\mathcal{K}$, though explanatory performance is much worse. Without any external knowledge, the diagnostic accuracy is also inferior.[7] The deteriorated performance can be attributed to inconsistent wording of diagnosis names, which makes evaluation tough. High observational scores imply that observations in $R$ can be identified without relying on external knowledge. There can be some cues to spot them.

**Performance in individual domains.** Figure 5 summarizes the performance of LLama3 70B, GPT-3.5, and GPT-4 across different medical domains, evaluated using $Acc^{\text{diag}}$, $Obs^{\text{comp}}$ (Comp), and $Exp^{\text{all}}$ (Faith). Neurology gives the best diagnostic accuracy, where LLama3 achieved an accuracy of 0.779. GPT-4 also performed well (0.753). In terms of $Obs^{\text{comp}}$ and $Exp^{\text{all}}$, GPT-4's results were 0.437 and 0.280, respectively. However, GPT-4 yields a higher diagnostic accuracy score while a lower explanatory score, suggesting that the observations captured by the model or their rationalizations differ from human doctors.

**Diagnostic reasoning under conditions of incomplete observation.** In real-world scenarios, doctors often have to make diagnoses based on incomplete information. To explore this, we conducted experiments on the 73 amended cases which originally lack observation to the final diagnosis (refer to supplementary for detailed introduction of amended data point). One set of experiments used the unmodified original notes, labeled as "Original," while the other set used notes with added observations labeled as "Amended." We tested three models—Llama3 70B, GPT-3.5-turbo, and GPT-4 turbo—under two settings: one with only the procedural graph $\mathcal{G}$ and the other with the complete knowledge graph $\mathcal{K}$. The results are presented in Tables 5 and 6. We can observe that in both $\mathcal{G}$ and $\mathcal{K}$ settings, the performance on the Amended data was consistently better across all metrics compared to the Original data. This suggests that even a single added observation can significantly impact the model's diagnostic reasoning.

---

[7]We understand this comparison is unfair, as the prompts differ. We intend to give a rough idea about the challenge without external knowledge.

Table 5: Amendment ablation study using $\mathcal{G}$.

| Setting | Models | Diagnosis | | Observation | | | Explanation | |
|---|---|---|---|---|---|---|---|---|
| | | $Acc^{\text{cat}}$ | $Acc^{\text{diag}}$ | $Obs^{\text{pre}}$ | $Obs^{\text{rec}}$ | $Obs^{\text{comp}}$ | $Exp^{\text{com}}$ | $Exp^{\text{all}}$ |
| Original | LLama3 70B | 0.547 | 0.273 | $0.225_{\pm 0.143}$ | $0.472_{\pm 0.144}$ | $0.253_{\pm 0.138}$ | $0.216_{\pm 0.271}$ | $0.073_{\pm 0.087}$ |
| | GPT-3.5 turbo | 0.507 | 0.273 | $0.393_{\pm 0.216}$ | $0.355_{\pm 0.174}$ | $0.278_{\pm 0.151}$ | $0.207_{\pm 0.305}$ | $0.062_{\pm 0.093}$ |
| | GPT-4 turbo | 0.616 | 0.328 | $0.446_{\pm 0.211}$ | $0.418_{\pm 0.164}$ | $0.340_{\pm 0.178}$ | $0.242_{\pm 0.324}$ | $0.098_{\pm 0.137}$ |
| Amended | LLama3 70B | 0.698 | 0.534 | $0.250_{\pm 0.173}$ | $0.507_{\pm 0.134}$ | $0.240_{\pm 0.129}$ | $0.296_{\pm 0.354}$ | $0.133_{\pm 0.142}$ |
| | GPT-3.5 turbo | 0.671 | 0.411 | $0.487_{\pm 0.206}$ | $0.351_{\pm 0.152}$ | $0.310_{\pm 0.145}$ | $0.272_{\pm 0.321}$ | $0.092_{\pm 0.118}$ |
| | GPT-4 turbo | 0.726 | 0.547 | $0.546_{\pm 0.184}$ | $0.465_{\pm 0.148}$ | $0.412_{\pm 0.171}$ | $0.391_{\pm 0.374}$ | $0.180_{\pm 0.186}$ |

Table 6: Amendment ablation study using $\mathcal{K}$.

| Setting | Models | Diagnosis | | Observation | | | Explanation | |
|---|---|---|---|---|---|---|---|---|
| | | $Acc^{\text{cat}}$ | $Acc^{\text{diag}}$ | $Obs^{\text{pre}}$ | $Obs^{\text{rec}}$ | $Obs^{\text{comp}}$ | $Exp^{\text{com}}$ | $Exp^{\text{all}}$ |
| Original | LLama3 70B | 0.575 | 0.219 | $0.109_{\pm 0.233}$ | $0.443_{\pm 0.171}$ | $0.203_{\pm 0.186}$ | $0.304_{\pm 0.388}$ | $0.114_{\pm 0.135}$ |
| | GPT-3.5 turbo | 0.548 | 0.233 | $0.293_{\pm 0.243}$ | $0.218_{\pm 0.198}$ | $0.184_{\pm 0.166}$ | $0.251_{\pm 0.357}$ | $0.072_{\pm 0.106}$ |
| | GPT-4 turbo | 0.616 | 0.260 | $0.452_{\pm 0.241}$ | $0.410_{\pm 0.211}$ | $0.349_{\pm 0.223}$ | $0.467_{\pm 0.437}$ | $0.220_{\pm 0.256}$ |
| Amended | LLama3 70B | 0.685 | 0.537 | $0.261_{\pm 0.195}$ | $0.493_{\pm 0.230}$ | $0.277_{\pm 0.171}$ | $0.452_{\pm 0.407}$ | $0.185_{\pm 0.194}$ |
| | GPT-3.5 turbo | 0.657 | 0.465 | $0.390_{\pm 0.227}$ | $0.272_{\pm 0.194}$ | $0.232_{\pm 0.156}$ | $0.401_{\pm 0.394}$ | $0.127_{\pm 0.145}$ |
| | GPT-4 turbo | 0.712 | 0.589 | $0.534_{\pm 0.214}$ | $0.452_{\pm 0.180}$ | $0.401_{\pm 0.201}$ | $0.607_{\pm 0.442}$ | $0.286_{\pm 0.258}$ |

For Cardiology and Endocrinology, the diagnostic accuracy of the models is relatively low (GPT-4 achieved 0.458 and 0.468, respectively). Nevertheless, $Obs^{\text{comp}}$ and $Exp^{\text{all}}$ are relatively high. Endocrinology results in lower diagnostic accuracy and higher explanatory performance. A smaller gap may imply that in these two domains, successful predictions are associated with observations similar to those of human doctors, and the reasoning process may be analogous. Conversely, in Gastroenterology, higher $Acc^{\text{cat}}$) is accompanied by lower $Obs^{\text{comp}}$ and $Exp^{\text{all}}$ (especially for LLama3), potentially indicating a significant divergence in the reasoning process from human doctors. Overall, DiReCT demonstrates that the degree of alignment between the model's diagnostic reasoning ability and that of human doctors varies across different medical domains.

**Reliability of automatic evaluation.** We randomly pick out 100 samples from DiReCT and their prediction by GPT-4 over the task with $\mathcal{G}$ to assess the consistency of our automated metrics to evaluate the observational and explanatory performance in Section 3.3 to human judgments. Three physicians joined this experiment. For each prediction $\hat{o} \in \hat{\mathcal{O}}$, they are asked to find a similar observation in ground truth $\mathcal{O}$. For explanatory metrics, they verify if each prediction $\hat{z} \in \hat{\mathcal{E}}$ for $\hat{o} \in \hat{\mathcal{O}}$ align with ground-truth $z \in \mathcal{E}$ corresponding to $o$. A prediction and a ground truth are deemed aligned for both assessments if at least two specialists agree. We compare LLama3's and GPT-4's judgments to explore if there is a gap between these LLMs. As summarized in Table 7, GPT-4 achieves the best results, with LLama3 8B also displaying a similar performance. From these results, we argue that our automated evaluation metrics are consistent with human judgments, and LLama3 is sufficient for this evaluation, allowing the cost-efficient option. Detailed analysis is available in the supplementary material.

Table 7: Consistency of automated evaluation with human judgments. Evaluated by mean and confidence interval (CI).

| Model | Observation | | Rationalization | |
|---|---|---|---|---|
| | Mean | 95% CI | Mean | 95% CI |
| LLama3 8B | 0.887 | $0.844 \sim 0.878$ | 0.835 | $0.759 \sim 0.818$ |
| GPT-4 turbo | 0.902 | $0.830 \sim 0.863$ | 0.876 | $0.798 \sim 0.853$ |

**Prediction examples.** Figure 6 shows a sample generated by GPT-4. The ground-truth PDD of the input clinical note is `Hemorrhagic Stroke`. In this figure, purple, orange, and red indicate explanations only in the ground truth, only in prediction, and common in both, respectively; therefore, red is a successful prediction of an explanation, while purple and orange are a false negative and false positive. GPT-4 treats the observation of `aurosis fugax` as the criteria for diagnosing `Ischemic Stroke`. However, this observation only supports `Suspected Stroke`. Conversely, observation `thalamic hematoma`, which is the key indicator of `Hemorrhagic Stroke`, is regarded as a less important clue. Such observation-diagnosis correspondence errors lead to the model's misdiagnosis. In Figure 7, we can observe that GPT-4 can find the key observation for the diagnosis of GERD,

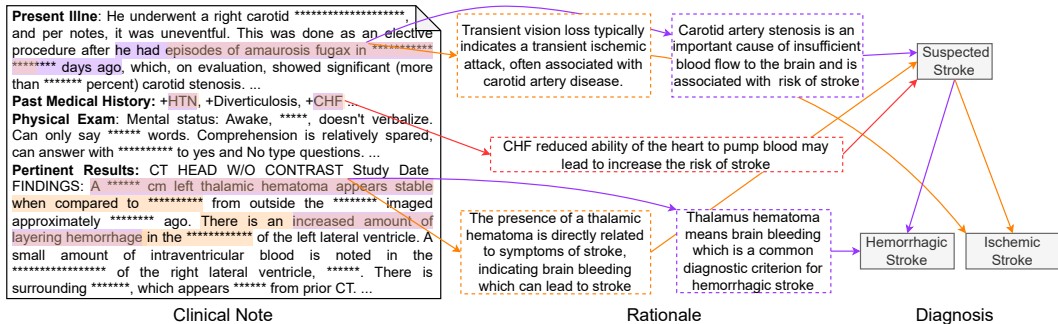

Figure 6: An example prediction for a clinical note with PDD of `Hemorrhagic Stroke` by GPT-4.

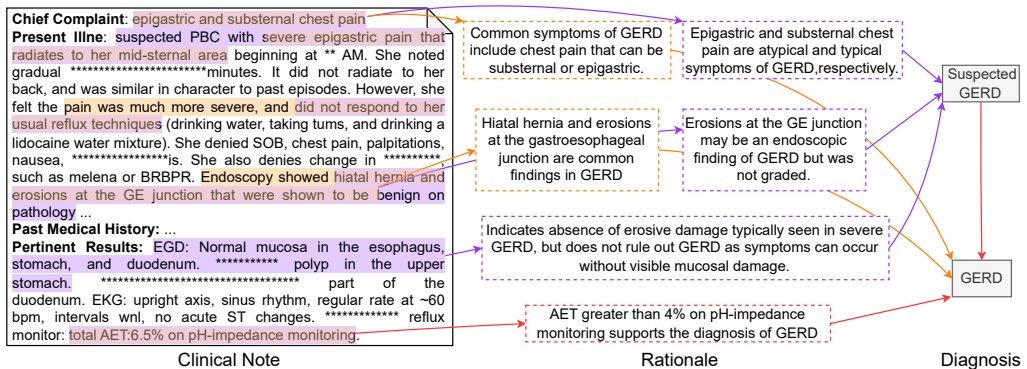

Figure 7: An example prediction for a clinical note with PDD of `GERD` by GPT-4

which is consistent with human in both observation and rationale. However, it still lacks the ability to identify all observations. More samples are available in the supplementary material.

## 6 Conclusion and Limitations

We proposed DiReCT as the first benchmark for evaluating the diagnostic reasoning ability of LLMs with interpretability by supplying external knowledge as a graph. Our evaluations reveal a notable disparity between current leading-edge LLMs and human experts, underscoring the urgent need for AI models that can perform reliable and interpretable reasoning in clinical environments. DiReCT can be easily extended to more challenging settings by removing the knowledge graph from the input, facilitating evaluations of future LLMs.

**Limitations.** DiReCT encompasses only a subset of disease categories and considers only one PDD, omitting the inter-diagnostic relationships due to their complexity—a significant challenge even for human doctors. Additionally, our baseline may not use optimal prompts or address issues related to hallucinations in task responses. Our dataset is solely intended for model evaluation but not for use in clinical environments. The use of the diagnostic knowledge graph is also limited to serving merely as a part of the input and once a knowledge graph is provided, the focus shifts to whether the LLM follows the graph's rules well (refer to supplementary). Future work will focus on constructing a more comprehensive disease dataset and developing an extensive diagnostic knowledge graph.

## Acknowledgments and Disclosure of Funding

This work was supported by World Premier International Research Center Initiative (WPI), MEXT, Japan. This work is also supported by JST ACT-X Grant Number JPMJAX24C8, JSPS KAKENHI No. 24K20795 and No. JP23H00497, CREST Grant No. JPMJCR20D3, JST FOREST Grant No. JPMJFR216O, and Dalian Haichuang Project for Advanced Talents.

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
