# Supplementary for DiReCT Paper

## A  Details of DiReCT

### A.1  Data Statistics

Table 1: Disease statistics of DiReCT.

| Domains | Categories | # samples | $|\mathcal{D}_i|$ | $|\mathcal{D}_i^\star|$ | References |
|---|---|---|---|---|---|
| Cardiology | Acute Coronary Syndromes | 65 | 6 | 3 | [Byrne et al., 2024, Kitaoka et al., 2020] |
| | Aortic Dissection | 14 | 3 | 2 | [Members et al., 2022] |
| | Atrial Fibrillation | 10 | 3 | 2 | [Joglar et al., 2024] |
| | Cardiomyopathy | 9 | 5 | 4 | [Ommen et al., 2020] |
| | Heart Failure | 52 | 6 | 3 | [Heidenreich et al., 2022] |
| | Hyperlipidemia | 2 | 2 | 1 | [Su et al., 2021, Mach et al., 2020] |
| | Hypertension | 32 | 2 | 1 | [Unger et al., 2020] |
| Gastroenterology | Gastritis | 27 | 5 | 3 | [Shah et al., 2021, of Gastroenterology et al., 2023, Banks et al., 2019, Chow et al., 2010] |
| | Gastroesophageal Reflux Disease | 41 | 2 | 1 | [Gyawali et al., 2024] |
| | Peptic Ulcer Disease | 28 | 3 | 2 | [Kavitt et al., 2019, Tarasconi et al., 2020] |
| | Upper Gastrointestinal Bleeding | 7 | 2 | 1 | [Barkun et al., 2019] |
| Neurology | Alzheimer | 10 | 2 | 1 | [McKhann et al., 1984] |
| | Epilepsy | 8 | 3 | 2 | [Igaku-Shoin-Ltd., 2018] |
| | Migraine | 4 | 3 | 2 | [Lipton et al., 2001, Eigenbrodt et al., 2021, Headache Classification Committee of the International Headache Society (IHS), 2018] |
| | Multiple Sclerosis | 27 | 6 | 4 | [Lublin, 2005, Brownlee et al., 2017] |
| | Stroke | 28 | 3 | 2 | [Kleindorfer et al., 2021] |
| Pulmonology | Asthma | 13 | 7 | 5 | [Qaseem et al., 2011, Bateman et al., 2007, Baos et al., 2018] |
| | COPD | 19 | 6 | 4 | [Gupta et al., 2013] |
| | Pneumonia | 20 | 4 | 2 | [Olson and Davis, 2020, RECOMMENDATIONS, 2012, Niederman et al., 2001] |
| | Pulmonary Embolism | 35 | 5 | 3 | [Konstantinides et al., 2020] |
| | Tuberculosis | 5 | 3 | 2 | [Lewinsohn et al., 2017] |
| Endocrinology | Adrenal Insufficiency | 20 | 4 | 3 | [Charmandari et al., 2014, Yanase et al., 2016, Bornstein et al., 2016] |
| | Diabetes | 13 | 4 | 2 | [ElSayed et al., 2023] |
| | Pituitary | 12 | 4 | 3 | [Tritos and Miller, 2023, Drummond et al., 2019, Cooper and Melmed, 2012, Mayson and Snyder, 2014] |
| | Thyroid Disease | 10 | 6 | 4 | [AlexanderErik et al., 2017] |

Table 1 provides a detailed breakdown of the disease categories included in DiReCT. The column labeled # samples indicates the number of data points. The symbols $|\mathcal{D}_i|$ and $|\mathcal{D}_i^\star|$ denote the total number of diagnoses (diseases) and PDDs, respectively. Existing guidelines for diagnosing diseases were used as References, forming the foundation for constructing the diagnostic knowledge graphs. As some premise may not included in the referred guidelines. During annotation, physicians will incorporate their own knowledge to complete the knowledge graph.

### A.2  Amended Data Points

Our proposed dataset aims to evaluate whether LLMs can provide a complete diagnostic reasoning process comparable to that of human doctors. To achieve this, we intended to select notes from the MIMIC database that contain comprehensive signs and symptoms as observations, enabling physicians to annotate the notes leading to a final PDD. For disease category like heart failure, MIMIC offers ample data, allowing us to choose notes with complete observations. However, for PDDs such as bacterial pneumonia, the number of relevant notes is limited, and many lack critical evidence necessary for diagnosis (e.g., sputum culture). We observed that in some notes, the section under the title 'sputum culture' was left blank. We suspect that this might be due to some information being missed in MIMIC. To annotate such cases, we ask physicians add the necessary observations to support the diagnosis. In total, we made amendments to 73 notes. These notes all lacked evidence for a final PDD diagnosis, and in each note, only one observation was added as evidence. Thus, the modifications to the original content of the notes were minimal. For example, in a note where the PDD is bacterial pneumonia, we only added the following description under 'pertinent results': 'Multiple organisms consistent with Haemophilus influenzae.'

To better illustrate the structure of our dataset and identify which data has been amended and what content has been added, we have provided a detailed CSV file on GitHub (https://github.com/wbw520/DiReCT/tree/master/utils/data_loading_analysisi). This file contains six columns, which record the following information: Disease Category, PDD, Data Root, Whether Amended, Amended Part, and Amended Content. The Data Root column records the path and filename of each note. We have stored the original note information and our annotations within a JSON file. The version submitted for review to PhysioNet follows the same storage format. In the Whether Amended column, notes that have been amended are marked as 'Yes,' with the Amended Part and Amended Content columns specifying which part of the note was modified and what content

| Notation | Description | Notation | Description |
|---|---|---|---|
| $R$ | The whole content of input note. | $U$ | The narrowing-down module. |
| $r$ | One data section of the input note. | $W$ | The perception module. |
| $\mathcal{D}$ | Disease collection. | $V$ | The reasoning module. |
| $\mathcal{D}^\star$ | PDD collection. | $\{d_n\}$ | Collection of children diagnosis. |
| $\mathcal{D}_i^\star$ | PDD collection for disease $i$. | $Acc^{diag}$ | Diagnosis accuracy for $d^\star$. |
| $d^\star$ | A PDD disease. | $Acc^{cat}$ | Diagnosis accuracy for category. |
| $d_t$ | Diagnosis at $t$-th iteration. | $Obs^{pre}$ | Precision of observation. |
| $d$ | A diagnosis in $\mathcal{G}$. | $Obs^{rec}$ | Recall of observation. |
| $\mathcal{G}$ | Procedural graphs. | $Obs^{comp}$ | Completeness of observation. |
| $g_i$ | Procedural subgraph for disease | $Exp^{com}$ | Completeness of explanation. |
| $\mathcal{K}$ | knowledge graphs. | $Exp^{all}$ | Completeness of all explanation. |
| $k_i$ | knowledge subgraphs for disease $i$. | $M$ | An language model. |
| $\mathcal{P}_i$ | Supporting edge collection for $k_i$. | $\mathcal{E}$ | Collection of annotated deductions. |
| $p$ | A premise defined in $K$. | | |
| $\mathcal{F}_i$ | Procedural edge collection for $g_i$. | | |
| $\mathcal{O}$ | Collection of annotated observation. | | |
| $o$ | An annotated observation. | | |
| $z$ | Rationale for a deduction. | | |
| $d_0$ | Root diagnosis for $g_i$. | | |

Table 2: Notations defined in this paper.

was added. Additionally, we have provided several synthetic annotated samples (non-MIMIC data) on GitHub, along with detailed instructions on the format of the annotated data and how to parse each JSON file.

### A.3 Structure of Knowledge Graph

We first show the notations definition on Table 2. The entire knowledge graph, denoted as $\mathcal{K}$, is stored in separate JSON files, each corresponding to a specific disease category $i$ as $\mathcal{K}_i$. Each $\mathcal{K}_i$ comprises a procedural graph $\mathcal{G}_i$ and the corresponding premise $p$ for each disease. As illustrated in Figure 1, the procedural graph $\mathcal{G}_i$ is stored under the key "Diagnostic" in a dictionary structure. A key with an empty list as its value indicates a leaf diagnostic node as $d^\star$. The premise for each disease is saved under the key of "Knowledge" with the corresponding disease name as an index. For all the root nodes (e.g., `Suspected Heart Failure`), we further divide the premise into "Risk Factors", "Symptoms", and "Signs". Note that each premise is separated by ";".

Our knowledge graph was directly constructed by human physicians who followed authoritative diagnostic guidelines and incorporated their clinical experience. For Cardiology, Gastroenterology, Neurology, Pulmonology, and Endocrinology, the knowledge graph was built by 2, 1, 2, 2, and 1 specialists from the respective departments. The construction process involved first defining the procedural graph $g_i$ for each category, followed by supplementing $g_i$ with the detailed premises corresponding to each diagnosis d to build $k_i$. The complete knowledge graphs are available on GitHub (https://github.com/wbw520/DiReCT/tree/master/utils/data_annotation).

### A.4 Annotation and Tools

We have developed proprietary software for annotation purposes. As depicted in Figure 2, annotators are presented with the original text as observations $o$ and are required to provide rationales ($z$) to explain why a particular observation $o$ supports a disease $d$. The left section of the figure, labeled Input1 to Input6, corresponds to different parts of the clinical note, specifically the chief complaint, history of present illness, past medical history, family history, physical exam, and pertinent results, respectively. Annotators will add the raw text into the first layer by left-clicking and dragging to select the original text, then right-clicking to add it. After each observation, a white box will be used to record the rationales. Finally, a connection will be made from each rationale to a disease, represented in a grey box. The annotation process strictly follow the knowledge graph. Both the final

{"Diagnostic":
    {"Suspected Heart Failure":
        {"Strongly Suspected Heart Failure":
            {"Heart Failure":
                {"HFrEF": [],
                "HFmrEF": [],
                "HFpEF": []}}}},
"Knowledge":
    {"Suspected Heart Failure":
        {"Risk Factors": "CAD; Hypertension; Valve disease; Arrhythmias; CMPs; Congenital heart disease; Infective; Drug-induced;
                Infiltrative, Storage disorders, Endomyocardial disease, Pericardial disease, Metabolic, Neuromuscular disease",
        "Symptoms": Breathlessness; Orthopnoea; Paroxysmal nocturnal dyspnoea; Reduced exercise tolerance; Fatigue; tiredness; increased
                time to recover after exercise; Ankle swelling; Nocturnal cough; Wheezing; Bloated feeling; Loss of appetite;
                Confusion (especially in the elderly); Depression; Palpitation; Dizziness; Syncope",
        "Signs": "Elevated jugular venous pressure; Hepatojugular reflux; Third heart sound (gallop rhythm); Laterally displaced apical
                impulse; Weight gain (>2 kg/week); Weight loss (in advanced HF); Tissue wasting (cachexia); Cardiac murmur; Peripheral
                edema (ankle, sacral, scrotal); Pulmonary crepitations; Pleural effusion; Tachycardia; Irregular pulse; Tachypnoea; Cheyne-Stokes
                respiration; Hepatomegaly; Ascites; Cold extremities; Oliguria;  Narrow pulse pressure."},
        "Strongly Suspected Heart Failure": "NT-proBNP > 125 pg/mL; BNP > 35 pg/mL",
        "Heart Failure": "Abnormal findings from echocardiography\uff1aLV mass index>95 g/m2 (Female), > 115 g/m2 (Male); Relative wall thickness >0.42;
                LA volume index>34 mL/m2; E/e ratio at rest >9; PA systolic pressure >35 mmHg; TR velocity at rest >2.8 m/s",
        "HFrEF": "LVEF<40%",
        "HFmrEF": "LVEF41-49%",
        "HFpEF": "LVEF>50%"}}

Figure 1: A sample of knowledge graph for `Heart Failure`. Each premise under the key of "Knowledge" is separated with ";".

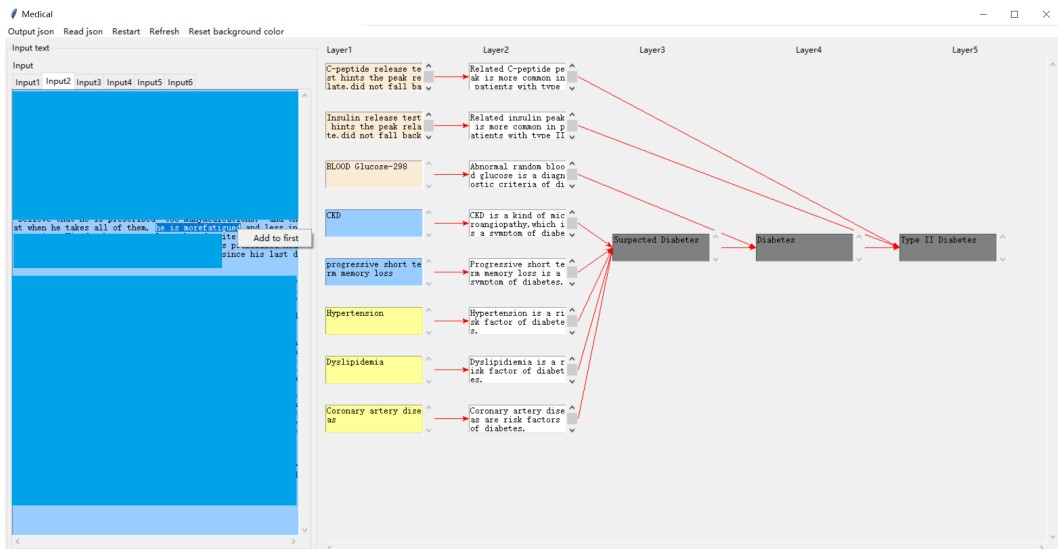

Figure 2: Demonstration of our annotation tool.

annotation and the raw clinical note will be saved in a JSON file. We provide the code to compile these annotations and detailed instructions for using our tool on GitHub.

## A.5   Access to DiReCT

Implementation code and annotation tool are available through https://github.com/wbw520/DiReCT. Data will be released through PhysioNet due to safety issues according to the license of MIMIC-IV (PhysioNet Credentialed Health Data License 1.5.0). We will use the same license for DiReCT. The download link will be accessible via GitHub. We confirm that this GitHub link and data link are always accessible. We confirm that we will bear all responsibility in case of violation of rights.

Table 3: Prompt for *narrowing-down* module.

| Input Prompt |
| --- |
| Suppose you are one of the greatest AI scientists and medical expert. Let us think step by step.
You will review a clinical 'Note' and your 'Response' is to diagnose the disease that the patient have for this admission.
All possible disease options are in a list structure: {disease_option}.
Note that you can only choose one disease from the disease options and directly output the origin name of that disease.
Now, start to complete your task.
Don't output any information other than your 'Response'.
'Note':
{note}
Your 'Response': |

Table 4: Prompt for *perception* module.

| Input Prompt |
| --- |
| Suppose you are one of the greatest AI scientists and medical expert. Let us think step by step.
You will review a part of clinical "Note" from a patient.
The disease for which the patient was admitted to hospital this time is {disease}.
Your task is to extract the original text as confidence "Observations" that lead to {disease}.
Here are some premise for the diagnosis of this disease category. You can refer them for your task. Premise are: {premise}
Note that you also need to briefly provide the "Reason" for your extraction.
Note that both "Observations" and "Reason" should be string.
Note that your "Response" should be a list structure as following
: [["Observation", "Reason"], ......, ["Observation", "Reason"]]
Note that if you can't find any "Observation" your "Response" should be: [].
Now, start to complete your task.
Note that you should not output any information other than your "Response".
"Note":
{note}
Note that you should not output any information other than your "Response".
Your "Response": |

## B  Implementation of Baseline Method

### B.1  Prompt Settings

In this section, we demonstrate the prompt we used for each module (From Table 3-5 for *narrowing-down*, *perception*, and *reasoning* module, respectively).

In Table 3, {disease_option} is the name for all disease categories, and {note} is the content for the whole clinical note. The response for the model is the name of a possible disease $\hat{i}$.

In Table 4, {disease} is the disease category name predicted in *narrowing-down*. The content marked blue is the premise, which is only provided during the $\mathcal{K}$ setting. In this module, {premise} is offered with all information in the knowledge graph. Different to *narrowing-down*, {note} is implemented for each clinical data $R = \{r\}$ and the outputs are combined together for $\hat{\mathcal{O}}$ and $\hat{\mathcal{E}}$.

In Table 5, {disease} is the disease category name and {disease_option} is consisted by the children nodes $\{d_n\}$. Similarly, the premise on the blue is only available for the $\mathcal{K}$ setting. It provides the premise that are criteria for the diagnosis of each children node. {observation} is the extracted $\hat{\mathcal{O}}$ in previous step. We provide all the prompts and the complete implementation code on GitHub.

### B.2  Diagnostic Reasoning Under Conditions of Incomplete Observation

In real-world scenarios, doctors often have to make diagnoses based on incomplete information. To explore this, we conducted experiments on the 73 amended cases. One set of experiments used the unmodified original notes, labeled as "Original," while the other set used notes with added observations, labeled as "Amended." We tested three models—Llama3 70B, GPT-3.5-turbo, and GPT-4 turbo—under two settings: one with only the procedural graph $\mathcal{G}$ and the other with the complete knowledge graph $\mathcal{K}$. The results are presented in Tables 6 and 7. We can observe that in both $\mathcal{G}$ and $\mathcal{K}$ settings, the performance on the Amended data was consistently better across all metrics compared to the Original data. This suggests that even a single added observation can significantly impact the model's diagnostic reasoning.

Table 5: Prompt for *reasoning* module.

| Input Prompt |
| --- |
| Suppose you are one of the greatest AI scientists and medical expert. Let us think step by step. |
| You will receive a list of "Observations" from a clinical "Note". These "Observations" are possible support to diagnose {disease}. |
| Based on these "Observations", you need to diagnose the "Disease" from the following options: {disease_option}. |
| Here are some golden standards to discriminate diseases. You can refer them for your task. Golden standards are: {premise} |
| Note that you can only choose one "Disease" from the disease options and directly output the name in disease options. |
| Note that you also required to select the "Observations" that satisfy the golden standard to diagnose the "Disease" you choose. |
| Note that you also required to provide the "Reason" for your choice. |
| Note that your "Response" should be a list structure as following |
| :[["Observation", "Reason", "Disease"], ......, ["Observation", "Reason", "Disease"]] |
| Note that if you can't find any "Observation" to support a disease option, your "Response" should be: None |
| Now, start to complete your task. |
| Note that you should not output any information other than your "Response". |
| "Observations": |
| {observation} |
| Note that you should not output any information other than your "Response". |
| Your "Response": |

Table 6: Amendment ablation study using $\mathcal{G}$.

| Setting | Models | Diagnosis | | Observation | | | Explanation | |
| --- | --- | --- | --- | --- | --- | --- | --- | --- |
| | | $Acc^{cat}$ | $Acc^{diag}$ | $Obs^{pre}$ | $Obs^{rec}$ | $Obs^{comp}$ | $Exp^{com}$ | $Exp^{all}$ |
| Original | LLama3 70B | 0.547 | 0.273 | $0.225_{\pm 0.143}$ | $0.472_{\pm 0.144}$ | $0.253_{\pm 0.138}$ | $0.216_{\pm 0.271}$ | $0.073_{\pm 0.087}$ |
| | GPT-3.5 turbo | 0.507 | 0.273 | $0.393_{\pm 0.216}$ | $0.355_{\pm 0.174}$ | $0.278_{\pm 0.151}$ | $0.207_{\pm 0.305}$ | $0.062_{\pm 0.093}$ |
| | GPT-4 turbo | 0.616 | 0.328 | $0.446_{\pm 0.211}$ | $0.418_{\pm 0.164}$ | $0.340_{\pm 0.178}$ | $0.242_{\pm 0.324}$ | $0.098_{\pm 0.137}$ |
| Amended | LLama3 70B | 0.698 | 0.534 | $0.250_{\pm 0.173}$ | $0.507_{\pm 0.134}$ | $0.240_{\pm 0.129}$ | $0.296_{\pm 0.354}$ | $0.133_{\pm 0.142}$ |
| | GPT-3.5 turbo | 0.671 | 0.411 | $0.487_{\pm 0.206}$ | $0.351_{\pm 0.152}$ | $0.310_{\pm 0.145}$ | $0.272_{\pm 0.321}$ | $0.092_{\pm 0.118}$ |
| | GPT-4 turbo | 0.726 | 0.547 | $0.546_{\pm 0.184}$ | $0.465_{\pm 0.148}$ | $0.412_{\pm 0.171}$ | $0.391_{\pm 0.374}$ | $0.180_{\pm 0.186}$ |

Additionally, we found that under the Amended data, using $\mathcal{K}$ led to both better diagnostic outcomes and improved explanability, aligning with the analysis in our paper. However, when using $\mathcal{K}$ on the Original data, while explanability improved, diagnostic accuracy actually decreased.

We conducted a detailed analysis of the 73 Original data's results from GPT-4. We found that GPT-4 was still able to correctly deduce the final PDD in 24 cases using $\mathcal{G}$ and 19 cases using $\mathcal{K}$. This indicates that the model possesses some level of uncertain reasoning capability. However, upon further inspection, we found that in some cases, the model used completely irrational observations as evidence, such as directly using "cough" as evidence for diagnosing "bacterial pneumonia". Additionally, there were 7 cases using $\mathcal{G}$ and 13 cases using $\mathcal{K}$ where the reasoning stopped before the final PDD diagnosis. This suggests that the model recognized the lack of sufficient evidence to derive the PDD and adhered faithfully to the diagnostic knowledge graph. Moreover, using appeared to help the model better understand this limitation, however, decrease the accuracy. These results indicate that employing the knowledge graph acts more like a trade-off: using only $\mathcal{G}$ results in a higher tendency for uncertain reasoning, while using the full $\mathcal{K}$ makes the model more cautious.

**Limitation of current implementation.** Once a knowledge graph is provided, the focus shifts to whether the LLM follows the graph's rules well. However, we consider the knowledge graph as an inferential framework rather than a set of rules. This framework provides decision-making paths for the LLM, but the LLM still needs to perform reasoning within it. Even when strictly following the knowledge graph, the LLM still needs to perform semantic analysis and context understanding in order to select the node that best suits the current situation among multiple possible paths (sub-nodes) in the knowledge graph. Therefore, the role of the LLM in this process is not merely to 'follow the rules,' but to make logical path selections based on its understanding of the input data, which itself is a reflection of reasoning ability. This often requires the model or algorithm to consider previous steps in each stage of reasoning and to update observations accordingly. Even revise or backtrack the diagnosis step. However, our baseline method did not account for this and thus cannot fully exploit this capability of the LLM, which is a current limitation. We did try some designs to give those abilities to LLMs, such as providing previous steps of reasoning for the current stage as input prompts or update observations. However, even GPT-4 cannot show high instruction following ability to realize them (maybe the input is too long or prompt setting problems).

Table 7: Amendment ablation study using $\mathcal{K}$.

| Setting | Models | Diagnosis | | Observation | | | Explanation | |
|---|---|---|---|---|---|---|---|---|
| | | $Acc^{\text{cat}}$ | $Acc^{\text{diag}}$ | $Obs^{\text{pre}}$ | $Obs^{\text{rec}}$ | $Obs^{\text{comp}}$ | $Exp^{\text{com}}$ | $Exp^{\text{all}}$ |
| Original | LLama3 70B | 0.575 | 0.219 | $0.109_{\pm 0.233}$ | $0.443_{\pm 0.171}$ | $0.203_{\pm 0.186}$ | $0.304_{\pm 0.388}$ | $0.114_{\pm 0.135}$ |
| | GPT-3.5 turbo | 0.548 | 0.233 | $0.293_{\pm 0.243}$ | $0.218_{\pm 0.198}$ | $0.184_{\pm 0.166}$ | $0.251_{\pm 0.357}$ | $0.072_{\pm 0.106}$ |
| | GPT-4 turbo | 0.616 | 0.260 | $0.452_{\pm 0.241}$ | $0.410_{\pm 0.211}$ | $0.349_{\pm 0.223}$ | $0.467_{\pm 0.437}$ | $0.220_{\pm 0.256}$ |
| Amended | LLama3 70B | 0.685 | 0.537 | $0.261_{\pm 0.195}$ | $0.493_{\pm 0.230}$ | $0.277_{\pm 0.171}$ | $0.452_{\pm 0.407}$ | $0.185_{\pm 0.194}$ |
| | GPT-3.5 turbo | 0.657 | 0.465 | $0.390_{\pm 0.227}$ | $0.272_{\pm 0.194}$ | $0.232_{\pm 0.156}$ | $0.401_{\pm 0.394}$ | $0.127_{\pm 0.145}$ |
| | GPT-4 turbo | 0.712 | 0.589 | $0.534_{\pm 0.214}$ | $0.452_{\pm 0.180}$ | $0.401_{\pm 0.201}$ | $0.607_{\pm 0.442}$ | $0.286_{\pm 0.258}$ |

Table 8: Prompt for evaluation of observation.

| Input Prompt |
|---|
| Suppose you are one of the greatest AI scientists and medical expert. Let us think step by step. You will receive two "Observations" extracted from a patient's clinical note. Your task is to discriminate whether they textually description is similar? Note that "Response" should be one selection from "Yes" or "No". Now, start to complete your task. Don't output any information other than your "Response". "Observation 1": {gt_observation} "Observation 2": {pred_observation} Your "Response": |

For evaluation, we used diagnostic processes annotated by human doctors as ground truth. Therefore, whether the KG is provided or not, the model's output needs to align with the ground truth. Our dataset allows for evaluation with and without the KG, but our baseline method is not effective at handling scenarios without the KG (this is much more challenging). How to utilize and explore the LLM's reasoning ability in this scenario is one of our future research directions.

## B.3 Details of Automatic Evaluation

The automatic evaluation is realized by LLama3 8B. We demonstrate the prompt for this implement in Table 8 (for observation) and Table 9 (for rationalization). Note that we do not use few-shot samples for the evaluation of observation. In Table 8, {gt_observation} and {pred_observation} are from model prediction and ground-truth. As this is a simple similarity comparison task to discriminate whether the model finds similar observations to humans, LLama3 itself have such ability. We do not strict to exactly match due to the difference in length of extracted raw text (as long as the observation expresses the same description). In Table 9, {gt_reasoning} and {pred_reasoning} are from model prediction and ground-truth. We require the rationale to be complete (content of the expression can be understood from the rationale alone) and meaningful; therefore, we provide five samples for this evaluation. We also provide all the prompts and the complete implementation code on GitHub.

For human evaluation, among the three specialists, two are from Cardiology and one is from Gastroenterology. Given that the notes originate from different medical domains, there is a possibility that the specialists may not be entirely accurate. However, this evaluation does not demand highly specialized knowledge, and it can be adequately covered by their expertise.

We also included an experimental result comparing the judgment differences between Llama3 8B and GPT-4 Turbo. The evaluation was performed on the diagnostic outcomes (across the entire dataset) from Llama3 70B and GPT-4 Turbo, using $\mathcal{G}$ as additional knowledge. We calculated the consistency rate for matching observations and the corresponding rationalization. As shown in Table 10, the differences in judgment between the two models are not obvious and are more consistent in observation discrimination. There are also some variations across different disease domains, with the highest similarity in observation discrimination found in Endocrinology, while the rationalization is most similar in Neurology.

Additionally, we provided results using GPT-4 Turbo for automatic evaluation, compared to those shown in Table 11 (which used Llama3 8B). The results indicate that GPT-4 Turbo tends to yield higher observation matching and more stringent rationalization discrimination. However, the largest difference does not exceed 5%. Considering the cost of GPT-4, Llama3 8B is a more efficient option.

Table 9: Prompt for evaluation of rationalization.

| Input Prompt |
|---|

Suppose you are one of the greatest AI scientists and medical expert. Let us think step by step.
You will receive two "Reasoning" for the explanation of why an observation cause a disease.
Your task is to discriminate whether they explain a similar medical diagnosis premise?
Note that "Response" should be one selection from "Yes" or "No".
Here are some samples:
Sample 1:
"Reasoning 1": Facial sagging is a classic symptom of stroke
"Reasoning 2": Indicates possible facial nerve palsy, a common symptom of stroke
"Response": Yes
Sample 2:
"Reasoning 1": Family history of Diabetes is an important factor
"Reasoning 2": Patient's mother had a history of Diabetes, indicating a possible genetic predisposition to stroke
"Response": Yes
Sample 3:
"Reasoning 1": headache is one of the common symptoms of HTN
"Reasoning 2": Possible symptom of HTN
"Response": No
Sample 4:
"Reasoning 1": Acute bleeding is one of the typical symptoms of hemorrhagic stroke
"Reasoning 2": The presence of high-density areas on Non-contrast CT Scan is a golden standard for Hemorrhagic Stroke
"Response": No
Sample 5:
"Reasoning 1": Loss of strength on one side of the body, especially when compared to the other side, is a common sign of stroke
"Reasoning 2": Supports ischemic stroke diagnosis
"Response": No
Now, start to complete your task.
Don't output any information other than your "Response".
"Reasoning 1": {gt_reasoning}
"Reasoning 2": {pred_reasoning}
Your "Response":

Table 10: Judgement consistency between LLama3 8B and GPT-4 turbo.

| Domain | LLama3 70B | | GPT-4 turbo | |
|---|---|---|---|---|
| | Observation | Rationalization | Observation | Rationalization |
| Cardiology | $0.885_{\pm 0.095}$ | $0.761_{\pm 0.268}$ | $0.827_{\pm 0.146}$ | $0.861_{\pm 0.273}$ |
| Gastroenterology | $0.862_{\pm 0.088}$ | $0.676_{\pm 0.361}$ | $0.810_{\pm 0.167}$ | $0.755_{\pm 0.316}$ |
| Neurology | $0.846_{\pm 0.090}$ | $0.831_{\pm 0.211}$ | $0.856_{\pm 0.124}$ | $0.963_{\pm 0.106}$ |
| Pulmonology | $0.808_{\pm 0.131}$ | $0.703_{\pm 0.317}$ | $0.786_{\pm 0.152}$ | $0.779_{\pm 0.287}$ |
| Endocrinology | $0.911_{\pm 0.104}$ | $0.783_{\pm 0.304}$ | $0.868_{\pm 0.145}$ | $0.793_{\pm 0.340}$ |
| Overall | $0.869_{\pm 0.102}$ | $0.734_{\pm 0.321}$ | $0.838_{\pm 0.144}$ | $0.806_{\pm 0.305}$ |

## B.4 Prediction Samples

Figure 3 and 4 shows two samples generated by GPT-4. The ground-truth PDD of the input clinical note is `Gastroesophageal Reflux Disease (GERD)` and `Heart Failure (HF)`. In these figures, purple, orange, and red indicate explanations only in the ground truth, only in prediction, and common in both, respectively; therefore, red is a successful prediction of an explanation, while purple and orange are a false negative and false positive.

In Figure 3, we can observe that GPT-4 can find the key observation for the diagnosis of GERD, which is consistent with human in both observation and rationale. However, it still lacks the ability to identify all observations and establish accurate relationships for diseases. In Figure 4, the model's predictions do not align well with those of a human doctor. Key observations, such as the relationships between BNP and LVEF, are incorrectly identified, leading to a final misdiagnosis.

## B.5 Experiments for No Extra Knowledge

We demonstrate the prompt used for $\mathcal{D}^\star$ and no knowledge settings in Table 12 and Table 13, respectively. {note} is the text of whole clinical note and {disease_options} in Table 12 is the name of all leaf node $\mathcal{D}^\star$.

Table 11: Result of using GPT-4 turbo and LLama3 8B for automatic evaluation.

| Judgement | Models | Observation | | | Explanation | |
|---|---|---|---|---|---|---|
| | | $Obs^{\text{pre}}$ | $Obs^{\text{rec}}$ | $Obs^{\text{comp}}$ | $Exp^{\text{com}}$ | $Exp^{\text{all}}$ |
| GPT-4 turbo | LLama3 70B | $0.317_{\pm 0.161}$ | $0.576_{\pm 0.195}$ | $0.294_{\pm 0.159}$ | $0.348_{\pm 0.300}$ | $0.107_{\pm 0.118}$ |
| | GPT-4 turbo | $0.465_{\pm 0.190}$ | $0.514_{\pm 0.157}$ | $0.408_{\pm 0.201}$ | $0.437_{\pm 0.335}$ | $0.187_{\pm 0.191}$ |
| LLama3 8B | LLama3 70B | $0.277_{\pm 0.146}$ | $0.537_{\pm 0.192}$ | $0.256_{\pm 0.142}$ | $0.395_{\pm 0.320}$ | $0.112_{\pm 0.110}$ |
| | GPT-4 turbo | $0.446_{\pm 0.207}$ | $0.491_{\pm 0.180}$ | $0.371_{\pm 0.186}$ | $0.475_{\pm 0.363}$ | $0.199_{\pm 0.181}$ |

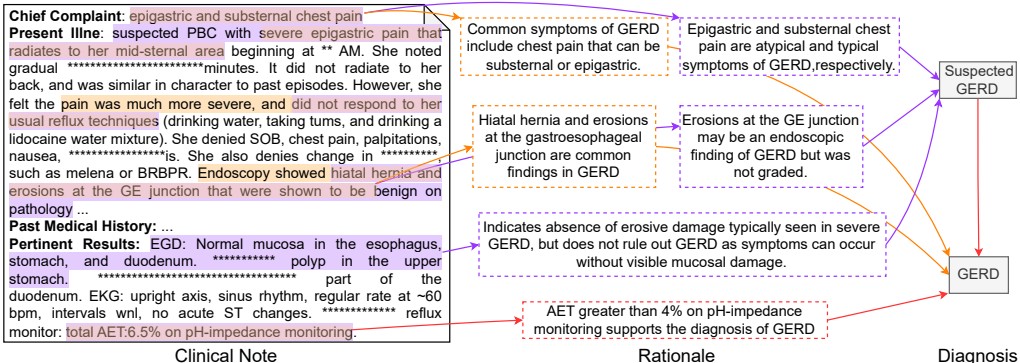

Figure 3: An example prediction for a clinical note with PDD of `GERD` by GPT-4

## B.6 Experimental Settings

All experiments are implemented with a temperature value of 0. All close sourced models are implemented in a local server with 4 NVIDIA A100 GPU.

## C Failed Attempts on DiReCT

In this section, we discuss some unsuccessful attempts during the experiments.

**Extract observation from the whole clinical note.** We try to diagnose the disease and extract observation, and the corresponding rationale using the prompt shown in Table 14. The {note} is offered by the whole content in the clinical note. We find that even though the model can make the correct diagnosis, only a few observations can be extracted (no more than 4), which decreases the completeness and faithfulness.

We also conducted an experiment to demonstrate the differences between two methods of observation extraction. The "Iteration" method is the one used in our paper, while the "Once" method is the one-time extraction method shown in Table 9. Each method was implemented under the condition of using GPT-4 turbo and Llama3 70B with $\mathcal{G}$ as input and was evaluated based on the Completeness of Observations (Obs) metric. The results are presented in Table 15. We found that while the "Once" extraction method resulted in higher precision, it led to a significant drop in recall, severely impacting the final completeness metric. The "Once" method tends to capture fewer observations, which hinders the overall reasoning process.

**End-to-End prediction.** We also try to output the whole reasoning process in one step (without iteration) when given observations. We show our prompt in Table 16. We find that using such a prompt model can not correctly recognize the relation between observation, rationale, and diagnosis.

## D Ethical Considerations

Utilizing real-world EHRs, even in a de-identified form, poses inherent risks to patient privacy. Therefore, it is essential to implement rigorous data protection and privacy measures to safeguard sensitive information, in accordance with regulations such as HIPAA. We strictly adhere to the Data

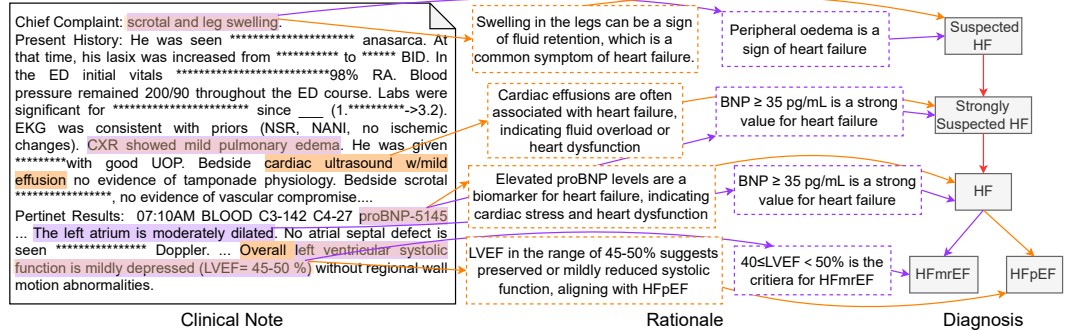

Figure 4: An example prediction for a clinical note with PDD of HF by GPT-4

Table 12: Prompt for $\mathcal{D}^\star$ setting.

| Input Prompt |
| --- |
| Suppose you are one of the greatest AI scientists and medical expert. Let us think step by step. |
| You will review a clinical 'Note' and your 'Response' is to diagnose the disease that the patient have for this admission. |
| All possible disease options are in a list structure: {disease_options}. |
| Note that you can only choose one disease from the disease options and directly output the origin name of that disease. |
| Now, start to complete your task. |
| Don't output any information other than your 'Response'. |
| 'Note': |
| {note} |
| Your 'Response': |

Use Agreement of the MIMIC dataset, ensuring that the data is not shared with any third parties. All experiments are implement on a private server. The access to GPT is also a private version.

AI models are susceptible to replicating and even intensifying the biases inherent in their training data. These biases, if not addressed, can have profound implications, particularly in sensitive domains such as healthcare. Unconscious biases in healthcare systems can result in significant disparities in the quality of care and health outcomes among different demographic groups. Therefore, it is imperative to rigorously examine AI models for potential biases and implement robust mechanisms for ongoing monitoring and evaluation. This involves analyzing the model's performance across various demographic groups, identifying any disparities, and making necessary adjustments to ensure equitable treatment for all. Continual vigilance and proactive measures are essential to mitigate the risk of biased decision-making and to uphold the principles of fairness and justice in AI-driven healthcare solutions.

Table 13: Prompt for no knowledge setting.

| Input Prompt |
| --- |
| Suppose you are one of the greatest AI scientists and medical expert. Let us think step by step. |
| You will review a clinical 'Note' and your 'Response' is to diagnose the disease that the patient have for this admission. |
| Note that you can only give one disease name and directly output the name of that "Disease". |
| Now, start to complete your task. |
| Don't output any information other than your 'Response'. |
| 'Note': |
| {note} |
| Your 'Response': |

Table 14: Prompt for extracting observation in one step.

| Input Prompt |
| --- |
| Suppose you are one of the greatest AI scientists and medical expert. Let us think step by step. |
| You will review a clinical 'Note', and your 'Response' is to diagnose the disease that the patient has for this admission. |
| All possible disease options are in a list structure: {disease_options}. |
| Note that you can only choose one disease from the disease options and directly output the origin name of that disease. |
| Note that you also need to extract original text as confidence "Observations" that lead to the "Disease" you selected. |
| Note that you should extract all necessary "Observation". |
| Note that you also need to briefly provide the "Reason" for your extraction. |
| Note that both "Observations" and "Reason" should be string. |
| Note that your "Response" should be a list structure as following |
| :[["Observation", "Reason", "Disease"], ......, ["Observation", "Reason", "Disease"]] |
| Now, start to complete your task. |
| Don't output any information other than your 'Response'. |
| 'Note' |
| :{note} |
| Your 'Response': |

Table 15: Comparison for using Iteration and Once for observation extraction.

| Models | Iteration | | | Once | | |
|---|---|---|---|---|---|---|
| | $Obs^{pre}$ | $Obs^{rec}$ | $Obs^{comp}$ | $Obs^{pre}$ | $Obs^{rec}$ | $Obs^{comp}$ |
| LLama3 70B | $0.277_{\pm0.146}$ | $0.537_{\pm0.192}$ | $0.256_{\pm0.142}$ | $0.325_{\pm0.207}$ | $0.324_{\pm0.147}$ | $0.185_{\pm0.107}$ |
| GPT-4 turbo | $0.446_{\pm0.207}$ | $0.491_{\pm0.180}$ | $0.371_{\pm0.186}$ | $0.567_{\pm0.268}$ | $0.287_{\pm0.156}$ | $0.244_{\pm0.147}$ |

Table 16: Prompt for End-to-End prediction.

**Input Prompt**

Suppose you are one of the greatest AI scientists and medical expert. Let us think step by step.
You will receive a list of "Observations" from a clinical "Note" for the diagnosis of stroke.
Here is the diagnostic route of stroke in a tree structure:
-Suspected Stroke
      -Hemorrhagic Stroke
      -Ischemic Stroke
Here are some premise for the diagnosis of this disease. You can refer them for your task. Premise are: {premise}
Based on these "Observations", starting from the root disease, your target is to diagnose one of the leaf disease.
Note that you also required to provide the "Reason" for your reasoning.
Note that your "Response" should be a list structure as following
:[["Observation", "Reason", "Disease"], ......, ["Observation", "Reason", "Disease"]]
Note that if you can't find any "Observation" to support a disease option, your "Response" should be: None
Now, start to complete your task.
Note that you should not output any information other than your "Response".
"Observations":
{observation}
Note that you should not output any information other than your "Response".
Your "Response":