# OpenReview forum: "DiReCT: Diagnostic Reasoning for Clinical Notes via Large Language Models"
_NeurIPS.cc/2024/Datasets_and_Benchmarks_Track — NeurIPS 2024 Track Datasets and Benchmarks Poster_

### Official Review · Reviewer_B51N · 2024-07-10
**All good but some curious points are remained.**

**Rating:** 8
**Confidence:** 4
**Correctness:** None

**Review:**

1. I appreciate the methods for constructing the dataset and the tasks performed in this paper. However, I do not understand why the diagnosis is performed directly by the LLM. The diagnostic knowledge graph is already defined by clinicians for each diagnostic category. Shouldn't the LLM just link the entities from the note to the diagnostic knowledge graph, and then the diagnosis should be determined by following the graph? Definitely, for note-to-entity linking, the LLM needs to perform good observations(detection), and the observed information must be correctly linked to the entities. However, once the linking is done, there should be no further need for the LLM to make the diagnosis directly; it should just follow the graph to derive the diagnostic leaf. This step seems unnecessary for AI. In Figure 3, for example: Observation ("Scrotal and leg swelling") → Explanation ("Peripheral edema is a sign of heart failure") → Entity linking ("Suspected Heart Failure": {"Signs": Peripheral edema (ankle, sacral, scrotal);}) → Diagnosis ("Suspected HF"). Shouldn't “Explanation → Entity linking” needs to be added? In the proposed framework, diagnoses are made directly using explanations without strict entity linking to the diagnostic knowledge graph. Is there any intention here? If the diagnostic knowledge graph is strictly followed, an incorrect observation and entity linking would naturally lead to an incorrect diagnosis. The proposed framework appears to replace entity linking with the generation of explanations, which seems ambiguous. Why was this approach chosen? If the diagnostic knowledge graph is strictly followed, explanations could be optional, and accurately identifying which entity the observation links to would be more critical.


2. I have questions about the results in Table 4. The observations seem to perform well even in the "No knowledge" setting compared to Table 3. However, the performance of explanations drops significantly. This is also the case in the "With D*" setting. Comparing Table 3 and Table 4 seems crucial for analysis. Is the significant performance drop in explanation related to the evaluation method (by human)?


3. I am curious about the correlation between the performance of explanations and diagnosis. The proposed approach is the explanations rather than the entity linking. If diagnoses are determined by a rule-based algorithm (diagnostic knowledge graph) rather than by the LLM, poor note-to-entity linking would naturally lead to poor diagnosis performance. Analyzing the correlation of explanation performance on diagnosis performance could show how well explanations replace entity linking and how well the LLM reflects the pre-defined diagnostic knowledge graph. This is very interesting to me. Do you have any insights on this?


4. I wonder how the reasoning chain (diagnostic knowledge graph) was constructed. Was it derived from clinical textbooks or somewhere? Were there special considerations to make the reasoning chain suitable for MIMIC-IV notes? It seems challenging to apply it globally to all types of notes I guess. Was it designed considering the medical terminology primarily used in MIMIC-IV?


5. Is there a need to evaluate performance using a classifier-based encoder-only model? For example, could a model performing classification (e.g., fine-tuned BERT on MIMIC-IV notes) be used as a comparison baseline by taking MIMIC-IV notes as input and predicting the diagnosis? The test set could be the notes discussed in the paper. If the proposed framework outperforms such fine-tuned transformer models, it may indicate that simply providing LLMs with external knowledge yields can get better performance, potentially eliminating the need for fine-tuning BERT-based models on note-to-diagnosis tasks. Including such a setting as a benchmark might be beneficial.

**Strengths:**

- Developing an evaluation dataset with human (clinician) annotations for reasoning, which is a critical issue in the current field, is a significant achievement.
- The dataset and benchmark are highly relevant to real-world clinical scenarios.

**Additional Feedback:**

None

**Clarity:**

Yes, if the notations is added to figure, it might be really helpful for understanding.

**Documentation:**

Datasets will be released.

**Limitations:**

This dataset will likely be well-utilized as a benchmark to measure the diagnostic reasoning performance of LLMs. However, it is unclear what specific medical significance there is in using discharge notes to predict the primary discharge diagnosis (PDD).

**Opportunities For Improvement:**

1. Is the annotation provided for each note on the route (entity) followed in the knowledge graph to arrive at the final diagnosis? (I understand that observations, explanations, and diagnoses are provided for each note, but I am curious if the specific premise entities followed are also provided.)


2. Was there any process except the experiments presented in the paper to verify that the dataset is well-constructed?


3. The paper contains many notations, and most explanations are made using notations. In such cases, adding notations to figures would make it easier for readers to understand. I found it challenging to read due to the confusing notations. I hope the figures are more detailed and aligned with the notations.


4. Is there any result for Table 9? It makes sense that LLMs would perform observations iteratively like real-world clinician, but I am curious if there is a significant performance difference between making observations at once versus iteratively.


5. Did the physicians cover notes across all diagnostic categories, or were they assigned only to categories in their specialty?

**Relation To Prior Work:**

Good.

**Summary And Contributions:**

The paper "DiReCT: Diagnostic Reasoning for Clinical Notes via Large Language Models" presents a novel dataset and benchmark designed to evaluate the diagnostic reasoning capabilities of large language models (LLMs) in the healthcare domain. The dataset includes 521 clinical notes meticulously annotated by physicians and is accompanied by a diagnostic knowledge graph. The paper emphasizes the importance of interpretability and presents comprehensive evaluation metrics to assess the reasoning abilities of various LLMs.

I think this paper provides a valuable dataset for diagnostic reasoning, but I have a few questions, which I have included in the review below.

---

> ### Author Rebuttal · Authors · 2024-08-15
>
> We sincerely appreciate Reviewer B51N's time and effort in evaluating our paper, along with the valuable feedback provided. Below, we address each of the specific concerns raised and we hope they can solve your concerns.
>
> **Comment for Review**
>
> **1. The Choice for the Current Data Structure.**
>
> The issue raised by the reviewer was also considered during the dataset construction. However, we opted for our current construction approach and analysis method for several reasons:
>
> **(1)**
> If we were to solely focus on linking observations in the notes with premises from the knowledge graph, the task would essentially become similar to a Natural Language Inference (NLI) task. This has been explored in the study by Jullien et al. (2023) in NLI4CT, although their work utilized synthetic data with shorter inputs. However, such a data structure does not adequately assess the reasoning capabilities of LLMs; it merely searches for a matching relationship. While we acknowledge that strictly adhering to the knowledge graph might be a more straightforward diagnostic approach in some cases, our intention is to balance the structured information from the knowledge graph with the reasoning ability of LLMs to tackle more complex diagnostic scenarios. Our construction method is inspired by the entailment tree framework [Dalvi et al., 2021], where an observation is followed by an explanation that outlines the potential diseases it could lead to, akin to a chain-of-thought process, enabling the model to engage in reasoning. These explanations are meaningful because, in real-world applications, the diagnostic process often involves complex reasoning and contextual understanding. This requires grasping the potential dependencies between observations. Moreover, when observations are insufficient, strictly following the knowledge graph could hinder the diagnostic process, whereas leveraging LLMs' reasoning capabilities might yield plausible inferences. This approach is also more aligned with how physicians make diagnoses; while they adhere to existing diagnostic guidelines, they also analyze based on the actual situation and their experience.
>
> **(2)**
> The explanations we provide are written by physicians based on an observation and the corresponding premise. In a sense, these explanations can be equated to premises, albeit in a more flexible form. We believe this is reasonable because not all patients' observations are identical. Even if they lead to the same diagnosis, the explanations may differ. For example, "shortness of breath" and "shortness of breath for a week" both indicate the same premise, but their severity differs. Our design allows the LLM to further reason based on the observation and premise, thereby providing diagnostic results that are more consistent with the actual situation.
>
> Additionally, we will provide the premise from the knowledge graph corresponding to each observation as a supplementary part of the dataset. This allows for an evaluation of the model's performance on pure NLI tasks and its diagnostic accuracy when strictly adhering to the knowledge graph.
>
> **(3)**
> Moreover, these explanations help reveal the reasoning behind the LLMs’ decisions to associate an observation with a particular disease. On one hand, they highlight potential errors in the model’s reasoning. For instance, in our results, the LLMs associated "persistent cough" with "bacterial pneumonia," providing the explanation: "Persistent coughing may damage the airway mucosal barrier, making the lungs more susceptible to bacterial invasion and leading to the development of bacterial pneumonia." Although this association is incorrect, as cough alone cannot directly serve as a premise for bacterial pneumonia, such explanations aid in understanding the model’s motivations. On the other hand, for non-professional general public, these explanations are highly valuable as they help them comprehend the reasoning behind the diagnosis.
>
> **2. Significant Performance Drop in Explanations**
>
> We find good performance for observation, which we attribute to the strong observation extraction capabilities of LLMs, even in the absence of external knowledge. The reason for significant performance drop in Table 4 is that, even in the presence of $\mathcal{D}^*$, the model merely knows all possible types of primary discharge diagnosis (PDD) (as $\mathcal{D}^*$ encompasses all PDD categories). However, the diagnostic knowledge graph is not provided, leaving the model unable to follow the standard diagnostic reasoning logic. The model does not know how to associate observations with the diagnoses defined in G, nor does it understand the premise associated with each diagnosis.

---

> > ### Author Rebuttal · Authors · 2024-08-15
> >
> > **3. Correlation between Explanation and Diagnosis Performance**
> >
> > In our study, the diagnostic process primarily relies on the LLM to generate explanations, based on the rules of the diagnostic knowledge graph to assist in deriving the final diagnosis. Our main goal is to enhance the transparency and explainability of the diagnosis by providing rich explanations, rather than relying solely on entity linking. From the experimental results in Table 3, comparing the diagnostic and explanatory accuracy of GPT-4 in the cases of with $\mathcal{G}$ and with $\mathcal{K}$, we find that, despite only a 0.042 difference in the final diagnosis, the gap in explanatory performance is significant. This indicates that even without the premises in the knowledge graph, the model can still reach the final diagnosis, but it struggles to link observations with diagnoses effectively. However, after incorporating the full K, both the explanatory and diagnostic performance of the model improve, particularly with a 0.158 increase in explanation accuracy. This suggests that GPT-4 can effectively follow the diagnostic knowledge graph. Overall, explanatory performance and diagnostic performance are interrelated, though the improvement in explanation is more pronounced.
> >
> > We are currently unable to draw a definitive conclusion on whether explanations can completely replace entity linking, as we have not conducted related experiments. However, we will provide annotations that link premises with observations and explore the diagnostic performance under this framework.
> >
> > **4. Construction of the Diagnostic Knowledge Graph**
> >
> > Our knowledge graph was directly constructed by human doctors, who based their work on existing authoritative diagnostic guidelines (see Supplementary Material, Table 1) and their own clinical experience. The domains of Cardiology, Gastroenterology, Neurology, Pulmonology, and Endocrinology were each constructed by 2, 1, 2, 2, and 1 specialized doctors, respectively. During construction, the procedural subgraph $g_i$ was first established, and then the detailed premises associated with each diagnosis $d$ in $g_i$ were added to create subgraph $k_i$. The complete knowledge graph are available on GitHub (https://github.com/wbw520/DiReCT/tree/master/utils/data_annotation).
> >
> > We did not tailor the knowledge graph to the MIMIC dataset; instead, it was designed using standard medical terminology.
> >
> > **5. Evaluating Performance Using a Classifier-Based Encoder-Only Model**
> >
> > While it could be an interesting baseline to use a BERT model for classification, the purpose of our dataset is to explore the diagnostic reasoning capabilities of LLMs. Therefore, standalone classification performance evaluation does not align with our objectives. Additionally, training a BERT-based classifier would require annotated data beyond pre-training, and given the limited amount of data number of our dataset, training a classifier could be quite challenging.
> >
> > **Comment for Opportunities For Improvement**
> >
> > **1. Annotation for Specific Premise Entities**
> >
> > We will also provide the associations of premises from the knowledge graph corresponding to each observation as a supplementary part of the dataset. This will allow us to evaluate diagnostic performance under the condition of strictly adhering to the knowledge graph.
> >
> > **2. Process to Verify Dataset Construction**
> >
> > To better demonstrate the structure of our dataset, we have made available a CSV file on GitHub (https://github.com/wbw520/DiReCT/tree/master/utils/data_loading_analysisi) that records detailed data information, including disease types for each note and storage root. This file follows the same storage format as the version submitted for review to PhysioNet. Additionally, we have provided several synthetic annotation samples (non-MIMIC data) on GitHub, along with detailed documentation explaining the format of our annotated data, the annotation process, and the method for parsing each annotated note.
> >
> > **3. Notations and Figures**
> >
> > We have added the relevant notations to Figures 3 and 4 to facilitate understanding. Please refer to the attached PDF file for details.
> >
> > **4. Results for Table 9**
> >
> > We conducted an experiment to demonstrate the differences between two methods of observation extraction. The "Iteration" method is the one used in our paper, while the "Once" method is the one-time extraction method shown in Table 9. Each method was implemented under the condition of using GPT-4 turbo and Llama3 70B with $\mathcal{G}$ as input and was evaluated based on the Completeness of Observations (Obs) metric. The results are presented in Table 1 of the attached PDF file. We found that while the "Once" extraction method resulted in higher precision, it led to a significant drop in recall, severely impacting the final completeness metric. The "Once" method tends to capture fewer observations, which hinders the overall reasoning process.
> >
> > **5. Physicians for Annotation**
> >
> > During the annotation process, notes from different domains were annotated by physicians specialized in those areas. Specifically, Cardiology, Gastroenterology, Neurology, Pulmonology, and Endocrinology were annotated by 3, 2, 1, 2, and 1 physicians, respectively.
> >
> > **Limitations: Predict the Primary Discharge Diagnosis (PDD)**
> >
> > In our dataset, each note only records admission information, as discharge checking and treatment plans in MIMIC have been removed. Therefore, our dataset simulates the process of diagnosing potential diseases at the time of a patient's admission based on the available information. The PDD, on the other hand, represents the ground truth disease diagnosed for the patient as documented in the note.

---

> > > ### Comment · Reviewer_B51N · 2024-08-25
> > >
> > > Thank you very much for your kind response. I believe your research will serve as an excellent milestone.
> > >
> > > However, I still have some questions regarding your answers.
> > >
> > > (1) "While we acknowledge that strictly adhering to the knowledge graph might be a more straightforward diagnostic approach in some cases, our intention is to balance the structured information from the knowledge graph with the reasoning ability of LLMs to tackle more complex diagnostic scenarios."
> > >
> > > -> If you're strictly adhering to the knowledge graph, wouldn't that mean the LLM doesn't really need to determine the next step in the diagnostic graph? It just needs to follow the graph to move to the next step.
> > > While I agree that reasoning can involve explanation, I'm not sure if moving the diagnostic graph to the next node based on LLM's decision can truly be considered as leveraging its reasoning capability.
> > > I understand that the graph was defined for the purpose of evaluation, but this seems to limit what could be considered as reasoning in the LLM.
> > > I’m uncertain about how much reasoning is involved in simply moving to the next node of the graph under strictly given conditions. For example, in your experiments with "No knowledge" or "Only giving D*," the LLM's output might genuinely reflect reasoning through the explanations it provides (Not strictly following knoweldge grpah because in those settings we don't provide KG to LLM).
> > > However, once a knowledge graph is provided, the focus shifts to whether the LLM follows the graph's rules well, making it difficult to argue that this demonstrates its reasoning capability.
> > >
> > > (2) "Moreover, when observations are insufficient, strictly following the knowledge graph could hinder the diagnostic process, whereas leveraging LLMs' reasoning capabilities might yield plausible inferences"
> > >
> > > -> When observations are insufficient, as you said, it’s impossible to strictly follow the knowledge graph (I agree). But doesn't it also seem same case to the LLM (can still perform reasoning based on insufficient observations?).
> > > Do you have any evidence to support your claim that the LLM's reasoning capabilities can be leveraged in situations with insufficient observations? (Any experiments for this?)
> > >
> > > (3) "Our construction method is inspired by the entailment tree framework [Dalvi et al., 2021], where an observation is followed by an explanation that outlines the potential diseases it could lead to, akin to a chain-of-thought process, enabling the model to engage in reasoning"
> > >
> > >  -> In your work, the process of identifying observations (perception) is separate from reasoning iteration, isn’t it? During reasoning iterations, you don't revisit or provide feedback on the observations; The framework keeps "given observation" by the perception process. So, you’re talking about something different here I guess.
> > >
> > > (4) "Moreover, these explanations help reveal the reasoning behind the LLMs’ decisions to associate an observation with a particular disease"
> > >
> > > -> Do you have experimental results to support this?
> > > Is there any evidence that the LLM's decision-making performance improved because of the explanations provided?
> > >
> > > Thank you.

---

> > ### Author Rebuttal · Authors · 2024-08-26
> >
> > Thank you very much for your discussion. Here are our responses to your questions:
> >
> > **1 Can the reasoning ability of LLM be truly reflected if the knowledge graph is strictly followed?**
> >
> > We agree with your view that “once a knowledge graph is provided, the focus shifts to whether the LLM follows the graph's rules well.” However, we consider the knowledge graph as an inferential framework rather than a set of rules. This framework provides decision-making paths for the LLM, but the LLM still needs to perform reasoning within it. Even when strictly following the knowledge graph, the LLM still needs to perform semantic analysis and context understanding in order to select the node that best suits the current situation among multiple possible paths (sub-nodes) in the knowledge graph. Therefore, the role of the LLM in this process is not merely to 'follow the rules,' but to make logical path selections based on its understanding of the input data, which itself is a reflection of reasoning ability. This often requires the model or algorithm to consider previous steps in each stage of reasoning and to update observations accordingly. Even revise or backtrack the diagnosis step. However, our baseline method did not account for this and thus cannot fully exploit this capability of the LLM, which is a current limitation. We did try some designs to give those abilities to LLMs, such as providing previous steps of reasoning for the current stage as input prompts or update observations. However, even GPT-4 cannot show high instruction following ability to realize them (maybe the input is too long or prompt setting problems).
> >
> > For evaluation, we used diagnostic processes annotated by human doctors as ground truth. Therefore, whether the KG is provided or not, the model's output needs to align with the ground truth. Our dataset allows for evaluation with and without the KG, but our baseline method is not effective at handling scenarios without the KG (this is much more challenging). How to utilize and explore the LLM's reasoning ability in this scenario is one of our future research directions.
> >
> > **2 Support for the reasoning capabilities where observations are insufficient.**
> >
> > We implemented an experiment to prove this capability. In our annotation, we observed that in some notes, there is a lack of critical evidence necessary for PDD. We suspect that this might be due to some information being missed in MIMIC. To annotate such cases, we ask physicians to add the necessary observations to support the diagnosis. In total, we made amendments to 73 notes (details shown in https://github.com/wbw520/DiReCT/tree/master/utils/data_loading_analysisi). These notes all lacked evidence for a final PDD diagnosis, and in each note, only one observation was added as evidence.  Thus, using these amended notes and original notes, we can explore how LLMs perform (whether they still diagnose PDD without key observation).
> >
> > We conducted experiments on the above 73 cases. One set of experiments used the unmodified original notes, labeled as "Original," while the other set used notes with added observations, labeled as "Amended." We tested three models under two settings: one with only the procedural graph $\mathcal{G}$ and the other with the complete knowledge graph $ \mathcal{K}$. The results are presented in Tables 1 and 2 of the attached PDF. We observed that in both $ \mathcal{G}$ and $ \mathcal{K}$ settings, the performance on the Amended data was consistently better across all metrics compared to the Original data. This suggests that even a single added observation can significantly impact the model’s diagnostic reasoning.
> >
> > To understand this better, we conducted a detailed analysis of the 73 Original data results from GPT-4. We found that GPT-4 was still able to correctly deduce the final PDD in 24 cases using $ \mathcal{G}$ and 19 cases using $ \mathcal{K}$. This indicates that the model may possess uncertain reasoning capability. For example, in one case, LLMs associated "persistent cough" with "bacterial pneumonia," providing the explanation: "Persistent coughing may damage the airway mucosal barrier, making the lungs more susceptible to bacterial invasion and leading to the development of bacterial pneumonia." Although it is not the premise for "bacterial pneumonia", it is the most important observation for diagnosing PDD in such an “incomplete” situation. Moreover, an explanation aids humans in understanding the model’s motivations is necessary, especially for such an uncertain reasoning.
> > Additionally, there were 7 cases using $ \mathcal{G}$ and 13 cases using $ \mathcal{K}$ where the reasoning stopped before the final PDD diagnosis. This suggests that the model recognized the lack of sufficient evidence to derive the PDD and adhered faithfully to the diagnostic knowledge graph. Moreover, using $ \mathcal{K}$ appeared to help the model better understand this limitation, however, decrease the accuracy. These results indicate that employing the knowledge graph acts more like a trade-off: using only $ \mathcal{G}$ results in a higher tendency for uncertain reasoning, while using the full $ \mathcal{K}$ makes the model more cautious.
> >
> > **3 Construction of our dataset.**
> >
> > In this sentence, we aim to mention that the data structure (annotation) is similar to the entailment tree. The phrase 'engage in reasoning' does not accurately convey our intended meaning of 'explanation.' For our baseline method, perception and reasoning are separated. As we discussed in comment 1, this is the limitation.
> >
> > **4 Explanation for decision-making performance.**
> >
> > There may be some description errors here. For this point, we want to mention the explanation for the association between an observation to a diagnosis can help humans understand the behavior of LLMs. Just like the plain text explanation used in previous medical QA tasks. We currently lack evidence to prove that such explanations enhance performance.

---

> > > ### Comment · Reviewer_B51N · 2024-08-26
> > > **I really love your sincere response.**
> > >
> > > Thank you for your response. I am very pleased with your reply.
> > >
> > > I would appreciate it if you could clearly represent the limitations you agreed about my comments in the paper.
> > >
> > > Additionally, I think it would be beneficial to include the additional experiments results you conducted (the tables attached additional PDF) in the supplementary section and update the main text to reference these findings, particularly regarding the cases where the reasoning was halted before the final PDD diagnosis due to a lack of observations.
> > >
> > > This point seems really important and appears to provide very interesting insights for many other readers.
> > > I also want to applaud your enthusiasm in addressing this issue in a way that I could understand.
> > >
> > > Thank you for your great research. Your response has made me a fan of your work.
> > > (I also raised the score.)
> > >
> > > Thank you.

---

> > > > ### Author Response · Authors · 2024-08-27
> > > > **Thanks for reviewing our work.**
> > > >
> > > > Thank you very much for raising the score. We sincerely appreciate the time and effort you have devoted to reviewing our work. In the revised manuscript, we will clearly outline the limitations and include new experiments as supplementary material, which will also be referenced in the main text.

---

### Official Review · Reviewer_DFEz · 2024-07-25
**The dataset is good and quite useful, although its readability could be improved.**

**Rating:** 7
**Confidence:** 5
**Correctness:** Yes
**Clarity:** Yes

**Review:**

Pros: This paper introduces a solid data curation pipeline that focuses on clinically useful data for real-world diagnosis. Including human experts to outline the reasoning steps will significantly benefit the development of the medical AI community.

Cons: The main issue with the paper is its readability. The authors use numerous notations, which are acceptable but make the text challenging to navigate, especially since readers must search the context to find the definitions of each notation.

**Strengths:**

This paper introduces a solid data curation pipeline that focuses on clinically useful data for real-world diagnosis. Including human experts to outline the reasoning steps will significantly benefit the development of the medical AI community.

**Additional Feedback:**

No

**Documentation:**

Yes

**Ethics:**

No concerns.

**Opportunities For Improvement:**

Including a table of notations and their corresponding definitions would be beneficial. Additionally, better matching of notations and captions could help readers understand the content more directly. For example, in Figure 3, labeling the clinical note as 'R', the rationale as 'z', and the diagnosis as 'd' would enable readers to easily match each component in the figure with its corresponding notation.

**Relation To Prior Work:**

Yes

**Summary And Contributions:**

This paper constructs a diagnostic reasoning dataset (knowledge graphs) for clinical notes using MIMIC-IV and expert annotations to provide rationales. Such a dataset is particularly significant for the medical AI domain, as most datasets focus only on QA, NLI, or entity extraction, lacking detailed interpretability and real clinical utility. The authors use this dataset to benchmark large language models’ ability to read lengthy texts and identify necessary observations for multi-evidence entailment tree reasoning. The findings demonstrate that current models do not perform at the same level as human doctors. This dataset is especially valuable for evaluating domain-specific data, considering its real-world applicability and complexity.

---

> ### Author Rebuttal · Authors · 2024-08-15
>
> We sincerely appreciate Reviewer DFEz's time and effort in evaluating our paper, as well as the valuable feedback provided. Below, we have addressed the concerns raised, and we hope that our responses adequately resolve your concerns.
>
> **Clarify Notations Demonstration for Better Readability**
>
> Based on your feedback, we have added a table that presents all the notations and explains their meanings, which can be found in Table 1 of the supplementary PDF. This table aims to make the paper easier for readers to follow. Additionally, we have incorporated the corresponding notations into Figures 3 and 4 of the original manuscript to enhance reader comprehension and ensure alignment with the descriptions used in the text. The revised figures are shown as Figure 1 and Figure 2 in attached pdf.

---

### Official Review · Reviewer_kZFT · 2024-07-28

**Rating:** 7
**Confidence:** 4
**Correctness:** Yes.
**Clarity:** Yes.

**Review:**

**Strengths**:
1. The dataset provided rich annotations and could be valuable for evaluating the power/current development of LLMs in clinical decision-making and clinical reasoning. The work emphasizes the importance of interpretability in AI-driven medical applications, addressing a critical need for models that can provide understandable reasoning behind their diagnostic decisions.
2. The authors evaluate the reasoning capabilities of 7 recent LLMs from diverse families and model sizes. This assessment provides an overview of current LLM performance in clinical reasoning settings and offers insights for future directions.
3. The inclusion of a diagnostic knowledge graph based on established guidelines enhances the dataset's utility by supplying essential knowledge for reasoning, which may not be covered in the training data of existing models. This supports more consistent annotations and aids in the interpretability of the models.


**Weaknesses:**
1. The dataset is currently under review for PhysioNet, which is a positive step towards addressing ethical concerns. However, since the raw data is not accessible, it is difficult to fully evaluate the dataset without seeing the samples.

**Strengths:**

Please see the review section above.

**Additional Feedback:**

N/A

**Documentation:**

Yes.

**Ethics:**

I believe the authors have addressed ethical issues effectively. They managed the ethical concerns of using MIMIC data for APIs by utilizing Microsoft Azure and are handling the ethics of publishing their dataset through PhysioNet.

**Limitations:**

Please see the review section above.

**Opportunities For Improvement:**

Please see the review section above.

**Relation To Prior Work:**

Yes.

**Summary And Contributions:**

The paper presents DiReCT (Diagnostic Reasoning for Clinical Notes via Large Language Models), a dataset designed to enhance the diagnostic reasoning capabilities of large language models (LLMs) in clinical settings. It consists of 521 domain expert-annotated clinical notes that detail the diagnostic process, linking observations to diagnoses. The dataset incorporates a diagnostic knowledge graph based on established guidelines to improve consistency in annotations and support reasoning that may not be covered in the model's training data.

---

> ### Author Rebuttal · Authors · 2024-08-15
>
> We sincerely appreciate the time and effort Reviewer kZFT has dedicated to evaluating our paper, as well as the valuable feedback provided. Below, we have addressed the concern raised and hope that our responses satisfactorily address your concern.
>
> **Access To Dataset**
>
> Our dataset is now under reviewing on PhysioNet. At this stage, to better demonstrate the structure of our dataset, we have made available a CSV file on GitHub (https://github.com/wbw520/DiReCT/tree/master/utils/data_loading_analysisi) that records detailed data information, including disease types for each note and storage root. This file follows the same storage format as the version submitted for review to PhysioNet. Additionally, we have provided several synthetic annotation samples (non-MIMIC data) on GitHub, along with detailed documentation explaining the format of our annotated data, the annotation process, and the method for parsing each annotated note. Once the review is approved, users can immediately utilize our data for their own experiments.

---

> > ### Comment · Reviewer_kZFT · 2024-08-19
> > **Reply to rebuttal**
> >
> > I have reviewed the rebuttal and appreciate the authors' responses to my question.
> > I look forward to the public release of the dataset.

---

> > > ### Author Response · Authors · 2024-08-20
> > > **Thank you**
> > >
> > > We sincerely appreciate your time and effort in reviewing our paper.

---

### Official Review · Reviewer_54eB · 2024-07-29
**An interesting paper, but a number of questions about the data construction and it's validity**

**Rating:** 7
**Confidence:** 4
**Clarity:** The paper is generally well written.

**Review:**

This is an interesting, and timely paper. It is generally well written, and contributes a benchmark that is much needed for the medical LLM domain. The authors additionally perform a comprehensive evaluation of both closed- and open-source models on the diagnostic capability flow with their proposed dataset.
The results provide an interesting baseline for future work.
However, there is a major retractor to this work, and a few minor areas of improvement that could solidify the paper.  The major weakness is the construction of the dataset: the authors note that annotators added information/“plausible clinical observations” to the note to match their diagnostic flow - however, clinicians often have to make diagnoses under relatively uncertain presentations. Additionally, authors do not assess model performance under this common-place clinical constraint, where information to make a diagnosis is missing. Furthermore, the authors rely on knowledge graphs to  perform this benchmark. However, the construction of the knowledge graphs (by whom, based on what resources, comparison to prior art), is not described. This weakens the dataset, as we cannot verify whether its construction is sound. This major weakness can be amended in two ways: 1) by providing a Datasheet For Datasets which details specifically how many datapoints were amended, the type of modifications, and dataset statistics,  as well as details on knowledge graph construction, and 2) performing an evaluation of the method under the “partial information” constraint that is common in clinical scenarios, and is more representative of the scenario. The minor areas concern the notation used throughout the paper, as well as the level of detail for the methods, which are further detailed below.

**Strengths:**

- **The paper tackles an important and interesting challenge for medical domain reasoning** This paper presents a timely benchmark for the evaluation of explainability and context attribution, knowledge usage, and diagnostic capability of models (primarily LLMs) in the medical domain. Most medical LLMs are evaluated on QA scenarios, which are not necessarily representative of a clinical workflow. Therefore, this dataset and benchmark presents a strong, relevant task to evaluate reasoning and specifically medical reasoning of LLMs.
- **The paper performs an comprehensive evaluation of currently available models on the provided dataset**

**Additional Feedback:**

-L23: “Despite the advancements, interpretability is critical, particularly in medical NLP tasks”, minor comment, consider rephrasing this paragraph to emphasise better *why* interpretability is critical in medical NLP. Furthermore, L24 “assess this capability over medical QA” - consider reword or explicitly explain the interpretability link to medical QA for these citations - whilst this is explained in the background, it would avoid confusion to clarify here.
    - L27: minor comment: consider linking two sentences (in real clinical settings can be more complex - as shown in Fig 1…), for better flow
    - L28: minor comment, consider reword: “a typical diagnosis requires comprehending and combining various information, such as health records, physical examinations, and laboratory tests, for further reasoning of possible diseases in a step-by-step manner following the established guidelines” - “a typical diagnosis requires comprehending and combining various sources of information… and laboratory tests, as well as reasoning over established guidelines over possible diseases  in a step-by-step manner.”
    - L53: “Multi-evidence entailment tree” - can the authors clarify if this is a novel contribution, or is this the of reasoning previously observed in the literature? Further in the text, Dalvi 2021 is proposed as a citation for the entailment tree method. Therefore, the authors should explicitly clarify within their contributions
    - L35: minor reword suggested, “the task consists of” instead of “the task basically is”
    - L118: Can the authors explain what is explicitly meant by “diagnostic flow”?
    - L174: Faithfulness of explanations Faith: typo?
    - Table 3 and 4: the titles of these tables should be reworked. They are too similar and thus slightly misleading/confusing. Additionally, the tables should be inserted into the text below their introduction, to facilitate reading of the results.

**Correctness:**

The comments on the dataset construction should be considered about correctness. Without further detail in a datasheet for datasets, and additionally without the dataset itself, it is hard to verify the soundness/correctness of the data.

**Documentation:**

There is no access to the dataset via a private URL. There is no datasheet for datasets to verify the construction of the dataset.

**Ethics:**

The authors state that they follow the Data Use Agreement for MIMIC-IV, therefore there are no ethical questions.

**Limitations:**

The work does not address the limitation of the dataset construction, see opportunities for improvement for more detail.

**Opportunities For Improvement:**

Firstly, here are some existing weaknesses in the paper which should be addressed:

- **The data is not fully representative of the clinical task, and it’s construction is not fully clear or described**: the authors state in footnote 3:  **“If R does not provide sufficient observations for the PDD (which may happen when a certain test is omitted), the annotators were asked to add plausible observations to R. This choice compromises the fidelity of our dataset to the original clinical notes, but we chose it for the completeness of the dataset.”**
This raises a number of important questions:

     - Firstly, how many datapoints were “amended” in this way? No statistics are provided on how many annotations underwent this process of “information completion”, and the data itself is not provided as it is not yet credentialed by PhysioNet.  This is significant to the understanding of the dataset - additional detail of the number of datapoints that were changed, how much information was added per note, and general dataset statistics must be released for a more comprehensive understanding of the dataset with respect to the original source, and it’s interpretation for future research.
     - Furthermore, these “additional” annotations raise the philosophical question of how the baseline method would work under the same constraints as typical clinicians. The authors should provide results on how these methods act under the typical clinical constraint of not having all signs and symptoms, or test results to confirm a diagnosis prior to making one by either ablating the current dataset and investigating the effects of partial information, or using the ‘unmodified” notes to this end and assess the difference in performance. This is a major detractor to the results, and therefore the authors should carefully consider this point.
      - L119: ”A disease category is defined according to an existing guideline, which starts from a certain diagnosis …” which existing guidelines are used? How were these graphs constructed? Please provide more detail on the construction and the knowledge graphs used for this benchmark in a data sheet for datasets.

- **The paper lacks notational clarity**: Generally the paper is well written, but the notation used throughout to define the graphs, baseline, and data is inconsistent and in places could use clarification. Specifically, the authors should consider reworking the explanation of their baseline section 4. Explicitly:
        - L111: could authors explicitly clarify the sentence “Our knowledge graphs $K = {K_i}_i$ is a collection of graph $K_i$ for disease category i., as $K_i$ is doubly indexed by I, as well as doubly noted for K? Consider: “Our knowledge graphs $\mathcal{K} = {k_i}$ is a set of graphs k, for disease category i.
        - L113: p ∈ Pi ; consider changing this to set notation as previously \mathcal{P_i} = { p}, similarly with d.
        - L115: where p ∈ Pi and d ∈ Di consider delete as this has previously been introduced
        - Eq 3): what is W? Also, presumably, The perception model can use G instead of K, as explained in 3.4. Consider here adding this.
        - L190: “Diagnosis dt identifies the set {dn}n of its children” what is n? Again, as previous feedback, why doubly index the set and the items?
        - L191: our reasoning module V - consider introducing this at the beginning  at line 184.
        - L195: What is t’? Additionally, can the authors clarify what is meant by “V is repeated until dt ′ in D⋆ ˆi is found or it fails”? What is failure in this context?
        - L196: “In our annotation, each observation contributes to deducing only one dt” - how did authors handle
        - L195: at this stage, it is slightly unclear how this algorithm works to me. Consider adding an algorithm to this section that explains the flow.

- **The paper is missing some details which have been moved to the supplementary which are critical to the understanding of the results and methods**: Generally, the paper is well detailed. However, in places, detail has been moved critically to the supplementary which is of importance to understand the results and methodology. Explicitly:
        - L183: “Figure 4 shows an overview of our baseline with three LLM-based modules narrowing-down,  perception, and reasoning (refer to the supplementary material for more details).” These are “non-standard“ LLM modules, or at least, are insufficiently described as to ascertain what type of LLM module this is. More detail needs to be provided for each of these here. Full details, like prompts, hyperparameters, etc, can be provided  in the supplementary materials, but otherwise it’s not possible to discern explicitly what each of these modules “do” and therefore how to interpret results in tables 3 + 4 against the method from the paper itself.


Secondly, here is a suggested extension to make the work more comprehensive:

 - **The paper would be bolstered by a more comprehensive evaluation of the automatic evaluation of “faithfulness”**: L258, the authors present a result for the reliability of the automatic evaluation of the “explanatory” performance of Llama3-8B as a judge.
        - How consistent are the clinical judges amongst them? How often do they disagree?
        - How consistent are the LLM judges (Llama3 8b GPT4) against each other? Does one succeed over another over certain diseases?
        - L267: the authors mention specialists: what are the clinicians’ specialties?
        - Table 5: more descriptive statistics are needed (IQR/confidence, statistical significance), if GPT-4 Turbo is statistically significantly better than Llama3, then ideally the tables 3 and 4 should be re-evaluated to assess difference in performance over the dataset, or at least a measure in the change of the results using  the LLM-as-a-judge should be provided to contextualise the table 3 and 4 results sensitivity to this tool.

**Relation To Prior Work:**

The paper is well contextualised with the prior literature, identifying a knowledge gap for medical LLM evaluation, making it a timely ad relevant contribution.

**Summary And Contributions:**

LLMs capacity to perform medical NLP tasks has often been evaluated in a QA setting. This paper proposes a novel benchmark, “Direct Reasoning for Clinical Notes”, DiReCt, which evaluates models’ abilities to follow diagnostic reasoning procedures, provide relevant explanations for diagnoses, and general diagnostic capability using clinical notes as their baseline.
The authors generate clinical diagnostic knowledge graphs G for a number of diseases, then annotate 521 clinical notes from the MIMIC-IV dataset, generating their diagnostic sub-knowledge graph K, extracting explanations for the diagnostic flow in order to provide attribution for the flow.
The paper then reports a benchmark a number of open- and closed-source models on various tasks: diagnostic accuracy (can a model diagnose correctly?), completeness of observations (can a model extract the correct number and correct type of supporting evidence for an inference?), and reasoning ability through evidence (is the flow towards diagnosis well supported by evidence, and well rationalised?).

---

> ### Author Rebuttal · Authors · 2024-08-15
>
> We sincerely appreciate the time Reviewer 54eB has dedicated to evaluating our paper, as well as the valuable feedback. Below, we address the specific concerns raised by the reviewer and we hope they can solve your concerns.
>
> **1. Representative of the Clinical Task and Dataset Construction Details**
>
> **1.1 About the Amended Data Points and Detailed Information**
>
> Our proposed dataset aims to evaluate whether LLMs can provide a complete diagnostic reasoning process comparable to that of human doctors. To achieve this, we intended to select notes from the MIMIC database that contain comprehensive signs and symptoms as observations, enabling physicians to annotate the notes leading to a final primary discharge diagnosis (PDD). For disease category like heart failure, MIMIC offers ample data, allowing us to choose notes with complete observations. However, for PDDs such as bacterial pneumonia, the number of relevant notes is limited, and many lack critical evidence necessary for diagnosis (e.g., sputum culture). We observed that in some notes, the section under the title 'sputum culture' was left blank. We suspect that this might be due to some information being missed in MIMIC. To annotate such cases, we ask physicians add the necessary observations to support the diagnosis. In total, we made amendments to 73 notes. These notes all lacked evidence for a final PDD diagnosis, and in each note, only one observation was added as evidence. Thus, the modifications to the original content of the notes were minimal. For example, in a note where the PDD is bacterial pneumonia, we only added the following description under 'pertinent results': 'Multiple organisms consistent with Haemophilus influenzae.'
>
> To better illustrate the structure of our dataset and identify which data has been amended and what content has been added, we have provided a detailed CSV file on GitHub (https://github.com/wbw520/DiReCT/tree/master/utils/data_loading_analysisi). This file contains six columns, which record the following information: Disease Category, PDD, Data Root, Whether Amended, Amended Part, and Amended Content. The Data Root column records the path and filename of each note. We have stored the original note information and our annotations within a JSON file. The version submitted for review to PhysioNet follows the same storage format. In the Whether Amended column, notes that have been amended are marked as 'Yes,' with the Amended Part and Amended Content columns specifying which part of the note was modified and what content was added. Additionally, we have provided several synthetic annotated samples (non-MIMIC data) on GitHub, along with detailed instructions on the format of the annotated data and how to parse each JSON file.
>
> **1.2 Baseline Method Under the Same Constraints as Typical Clinicians**
>
> We appreciate the reviewer’s insightful comments, particularly the notion that in real-world scenarios, doctors often have to make diagnoses based on incomplete information. However, this diverges slightly from our original intention. Our aim was to evaluate whether large language models (LLMs) can generate a complete diagnostic reasoning outcome, which assumes that the notes contain all necessary observational information. In clinical annotation, the absence of crucial evidence would make it impossible to derive a final PDD diagnosis. Therefore, our focus was on ensuring the annotations were both complete and accurate, without considering the uncertainties inherent in real-world medical diagnostics.
>
> Nonetheless, our dataset could be extended to test the performance of our baseline method in diagnostic reasoning under conditions of incomplete observation. To explore this, we conducted experiments on the 73 cases mentioned in our previous response. One set of experiments used the unmodified original notes, labeled as "Original," while the other set used notes with added observations, labeled as "Amended." We tested three models—Llama3 70B, GPT-3.5-turbo, and GPT-4 turbo—under two settings: one with only the procedural graph $\mathcal{G}$ and the other with the complete knowledge graph $ \mathcal{K}$. The results are presented in Tables 1 and 2 of the attached PDF. We observed that in both $ \mathcal{G}$  and $ \mathcal{K}$ settings, the performance on the Amended data was consistently better across all metrics compared to the Original data. This suggests that even a single added observation can significantly impact the model’s diagnostic reasoning.
>
> Additionally, we found that under the Amended data, using $ \mathcal{K}$ led to both better diagnostic outcomes and improved explanability, aligning with the analysis in our paper. However, when using $ \mathcal{K}$ on the Original data, while explanability improved, diagnostic accuracy actually decreased.
>
> We conducted a detailed analysis of the 73 Original data's results from GPT-4. We found that GPT-4 was still able to correctly deduce the final PDD in 24 cases using $ \mathcal{G}$  and 19 cases using $ \mathcal{K}$. This indicates that the model possesses some level of uncertain reasoning capability. However, upon further inspection, we found that in some cases, the model used completely irrational observations as evidence, such as directly using "cough" as evidence for diagnosing "bacterial pneumonia". Additionally, there were 7 cases using $ \mathcal{G}$  and 13 cases using $ \mathcal{K}$ where the reasoning stopped before the final PDD diagnosis. This suggests that the model recognized the lack of sufficient evidence to derive the PDD and adhered faithfully to the diagnostic knowledge graph. Moreover, using $ \mathcal{K}$ appeared to help the model better understand this limitation, however, decrease the accuracy. These results indicate that employing the knowledge graph acts more like a trade-off: using only $ \mathcal{G}$  results in a higher tendency for uncertain reasoning, while using the full $ \mathcal{K}$ makes the model more cautious.

---

> > ### Author Rebuttal · Authors · 2024-08-15
> >
> > **1.3 Construction of Knowledge Graph**
> >
> > We constructed the knowledge graph for two main reasons. First, existing large language models lack explicit diagnostic reasoning knowledge, making it difficult for them to align with human doctors. Second, previous knowledge graphs, such as the UMLS KG, only include relationships between disease entities and lack the ability to provide specific clinical decision support.
> >
> > Our knowledge graph was directly constructed by human physicians who followed authoritative diagnostic guidelines (refer to Supplementary Table 1) and incorporated their clinical experience. For Cardiology, Gastroenterology, Neurology, Pulmonology, and Endocrinology, the knowledge graph was built by 2, 1, 2, 2, and 1 specialists from the respective departments. The construction process involved first defining the procedural graph $g_i$ for each category, followed by supplementing $g_i$ with the detailed premises corresponding to each diagnosis d to build $k_i$ (we change the notation for subgraph to $g_i$ and $k_i$ according to the following comments). The complete knowledge graphs are available on GitHub (https://github.com/wbw520/DiReCT/tree/master/utils/data_annotation).
> >
> >
> > **2. Notational Clarity**
> >
> > **L111**: To improve distinction, we have defined knowledge graphs as $\mathcal{K}$= {$k_i$}, where $k_i$ represents the sub-graph for each disease category. We also defined the procedural graph as $ \mathcal{G}$ = {$g_i$} to maintain a consistent format.
> >
> > **L113**: We revised $p$ and $d$ with a consistent format as $ \mathcal{P}_i$ = {$p$}, $ \mathcal{D}_i$ = {$d$}.
> >
> > **L115**: The repeated description has been removed.
> >
> > **Equation 3**: $W$ is defined as the perception model. We have added this definition and included a description for the equation: "The perception model can also utilize $g_i$ instead of $k_i$ for the first task."
> >
> > **L190**: $n$ is the index of the children for $d_t$, and we have removed the double index.
> >
> > **L191**: A description was added in line 184: "$V$ iteratively derives possible diseases from observations based on the diagnosis knowledge graph, justifying each deduction $(o, z, d)$."
> >
> > **L195**: $t$ is the iteration index for the perception module $V$. Since $d_0$ is the root node of the procedural subgraph $g_i$ as calculated by the narrowing-down module $U$, the iteration for $V$ begins from $t = 1$.
> >
> > **L195**: In this context, "Failure" indicates that $V$ cannot identify any observation that supports diagnosing any child node in {$d_n$}, resulting in the termination of $V$'s iteration.
> >
> > **L196**: In our annotation, an observation $o$ is associated with only one $d$. However, our method employs an iterative reasoning pipeline. Initially, the perception module $W$ generates an explanation set $\mathcal{E}_0$, linking all $\hat{o}$ to $d_0$. During the $t$-th iteration of $V$, the explanation set is $\mathcal{E}_t$, where at least one $\hat{o}$ is linked to $d_t$. The final diagnosis explanation is the combination of {${\mathcal{E}_0, \dots, \mathcal{E}_T}$} and {${d_0, \dots, d_T}$}, where $T$ represents the final iteration. In this combination, if an $\hat{o}$ is eventually processed in the iteration for $\mathcal{E}_t$, the corresponding $(o, z, d)$ in all preceding {$\mathcal{E}_0$, $\dots$, $\mathcal{E}_t-1$} will be removed. That is $\hat{o}$ will always be possessed by the $d_t$ closest to leaf PDD node.
> >
> > **L195**: We found that the current explanation of our method was unclear, so we have clarified the reasoning module $V$ as follows: "After the perception module $W$ (iteration $t = 0$), we obtain all observations $\mathcal{\hat{O}}$, the root node of the diagnosis $d_0$, and an explanation $E_0$ for the initial iteration. Assuming that by iteration $t $, we already know the diagnosis for iteration $t$ as $d_t$. {$d_n$} is the set of $d_t$'s children, and ${\mathcal{P}_{\hat{i}}(d_n)}$
> >
> > represents the corresponding premises that support each $d_n$. $V$ identifies the diagnosis for the next step of $d_{t+1}$, and provides a justification $\mathcal{E}_{t+1}$.
> >
> > $V$ will verify if there is any $\hat{o}$ in $\mathcal{\hat{O}}$ that supports a $d_n$. If fully supported, $d_n$ is identified as $d_{t+1}$ for the $(t+1)$-th iteration, i.e.,
> >
> > $d_{t+1}, \mathcal{E}_{t+1}$ =  $V$($\hat{O}$ , {$d_n$},  {$\mathcal{P}_i$ ($d_n$)}).
> >
> > It continues until $d_{t+1}$ in $\mathcal{D}^*$ is identified. If no observation supports a $d_n$, the reasoning process will be stopped.” We then implement the combination mentioned in the previous comment (L196).
> >
> > **3. Missing Details**
> >
> > **L183**: We have provided more detailed descriptions of these modules to ensure that readers can clearly understand the functions and roles of each. Specifically, we have added explanations in the main text, covering their design concepts, operational logic, and specific roles in the experiments. The revised description is as follows: "Figure 4 provides an overview of our baseline, which comprises three LLM-based modules: narrowing-down ($U$), perception ($W$), and reasoning ($V$). In our experiments, each module utilizes the same type of LLM with different prompts (refer to the supplementary material for more details). $U$ analyzes the entire note $R$ to determine the possible disease type $\hat{i}$. $W$ extracts observations that may lead to diseases from each $r$, producing a list of original disease descriptions. $V$ iteratively derives possible diseases from observations based on the diagnosis knowledge graph, providing rationales for each deduction $(o, z, d)$."

---

> > ### Author Rebuttal · Authors · 2024-08-15
> >
> > **4. Comprehensive Evaluation of the Automatic Evaluation**
> > - L267: Among the three specialists, two are from Cardiology and one is from Gastroenterology. Given that the notes originate from different medical domains, there is a possibility that the specialists may not be entirely accurate. However, this evaluation does not demand highly specialized knowledge, and it can be adequately covered by their expertise.
> >
> > We also included an experimental result comparing the judgment differences between Llama3 8B and GPT-4 Turbo. The evaluation was performed on the diagnostic outcomes (across the entire dataset) from Llama3 70B and GPT-4 Turbo, using $\mathcal{G}$ as additional knowledge. We calculated the consistency rate for matching observations and the corresponding rationalization. As shown in Table 3 (attached in the PDF), the differences in judgment between the two models are not obvious, and are more consistent in observation discrimination. There are also some variations across different disease domains, with the highest similarity in observation discrimination found in Endocrinology, while the rationalization is most similar in Neurology.
> >
> > Additionally, we provided results using GPT-4 Turbo for automatic evaluation, compared to those shown in Table 4 (which used Llama3 8B). The results indicate that GPT-4 Turbo tends to yield higher observation matching and more stringent rationalization discrimination. However, the largest difference does not exceed 5%. Considering the cost of GPT-4, Llama3 8B is a more efficient option. We also included the 95% confidence interval for human evaluation, as shown in Table 5.
> >
> > **5. Additional Feedback**
> >
> > **L23**: We clarified the importance of interpretability in medical NLP with the following revision: “Despite technological advancements, interpretability is crucial in medical NLP tasks because these tasks directly impact patient health and treatment decisions. Without clear interpretability, there's a risk of misdiagnosis and improper treatment, making it vital for ensuring medical safety.”
> >
> > **L23**: To avoid confusion, we revised the statement as follows: “Some studies assess this capability in medical contexts using plain text explanations...”
> >
> > **L27**: We combined the sentences into one as follows: “Nevertheless, real-world clinical tasks are often more complex, as illustrated in Figure 1, ...”
> >
> > **L28**: We improved our writing as per your suggestions: “A typical diagnosis requires comprehending and integrating various sources of information, such as health records, physical examination, and laboratory tests, along with reasoning over established guidelines for potential diseases in a step-by-step manner.”
> >
> > **L53**: The entailment tree was proposed by Dalvi in 2021. However, this approach was designed for Natural Language Inference, where the premises for constructing such a tree are directly provided as input. In our work, the method needs to extract disease observations as evidence from a clinical note to construct the entailment tree. This task is significantly more complex, as multiple pieces of evidence must be extracted. We clarified our contribution as follows: “… multi-evidence entailment tree reasoning, an extension of the original entailment tree explanation for complex scenarios in medical NLP tasks. [Dalvi et al., 2021]”
> >
> > **L35**: We modified the description as follows: “The task involves predicting the diagnosis from ...”
> >
> > **L118**: The diagnostic flow here refers to the pathway from the root diagnosis to one of the PDDs. This is a step-by-step diagnostic process.
> >
> > **L174**: We removed the term “Faith” from this line, as it was intended to be an abbreviation but is not necessary.
> >
> > **Tables 3 and 4**: We renamed both tables as follows:
> >
> > Table 3: Evaluation of diagnostic reasoning ability using $\mathcal{G}$ or $\mathcal{K}$ as input.
> >
> > Table 4: Evaluation of diagnostic reasoning ability without external knowledge.

---

> > > ### Comment · Reviewer_54eB · 2024-08-23
> > >
> > > I thank the authors for their comprehensive rebuttal; I have accordingly updated my score to a 7.

---

> > > > ### Author Response · Authors · 2024-08-25
> > > > **Thank you**
> > > >
> > > > Thank you very much for raising the score. We appreciate your time and effort in reviewing our paper.

---

### Comment · Area_Chair_UiQm · 2024-08-29

I believe IRB approval or an IRB exemption is required for this study since it involves 12 human subjects for annotation and verification tasks and MIMIC-IV data was sent to HIPAA-compliant GPT-4. Moreover, it is unclear whether all human annotators had approved access to MIMIC-IV during the annotation process.

---

> ### Comment · Reviewer_B51N · 2024-08-30
>
> Dear AC,
>
> Hello, I'm asking this purely out of curiosity.
>
> I understand that participants involved in experiments to verify tasks require IRB approval, but do annotators also typically need IRB approval?
>
> In any case for this paper, I agree that IRB approval is necessary for this paper

---

> > ### Comment · Area_Chair_UiQm · 2024-08-30
> >
> > It is the responsibility of the IRB at each institution to determine whether full IRB approval is required or if the project qualifies for IRB exemption. However, the crucial point is that principal investigators must consult with their respective IRB prior to initiating any experiments. This ensures all research activities comply with ethical standards and regulatory requirements.

---

> ### Author Rebuttal · Authors · 2024-08-30
>
> Dear Area Chair,
>
> Thank you for your comment. We initially sought IRB approval for this research. However, our institution determined that a full IRB review was not required for this study because MIMIC-IV is open-source. Consequently, we were granted an IRB exemption. To further ensure the security of the data, all annotations were conducted on a secure, non-networked computer to prevent any possibility of data copying or transfer. Additionally, we confirm that all annotators have completed the required CITI training and have been granted the necessary legal access to the MIMIC-IV dataset.

---

### Decision · Program_Chairs · 2024-09-26

**Decision:**

Accept (Poster)

**Comment:**

The reviewers all appreciated the paper, which addresses an important challenge in diagnostic reasoning. The paper presents a robust data curation pipeline and timely benchmarking, which will significantly benefit the development of AI models in the medical domain. The evaluation is thorough and provides comprehensive comparisons of the LLMs, establishing solid baselines. The authors' response has clarified most of the points raised by the reviewers. In light of this, the authors are strongly encouraged to incorporate the feedback into the final version.